# CONFORMAL DATA CONTAMINATION TESTS FOR IN-DISTRIBUTION DATA ACQUISITION

## ABSTRACT

The amount of quality data in many machine learning tasks is limited to what is available locally to data owners. The set of quality data can be expanded through trading or sharing with external data agents. However, external data may be contaminated or introduce undesirable sample diversity which can degrade performance of personalized machine learning tasks, as in diagnosis of a rare disease or recommendation systems. Therefore, data buyers need quality guarantees prior to data acquisition. Previous works primarily rely on distributional assumptions about data from different agents, relegating quality checks to post-hoc steps involving costly data valuation procedures. We propose a distribution-free, contamination-aware data-sharing framework that, by inspecting only a small volume of data, identifies external data agents whose data is most valuable for model personalization. To achieve this, we introduce novel two-sample testing procedures, preceding full data acquisition, grounded in rigorous theoretical foundations for conformal outlier detection, to determine whether an agent's data exceeds a contamination threshold. The proposed tests, termed *conformal data contamination tests*, remain valid under arbitrary contamination levels while enabling false discovery rate control via the Benjamini-Hochberg procedure. Empirical evaluations across diverse collaborative learning scenarios demonstrate the robustness and effectiveness of our approach. Overall, the conformal data contamination test distinguishes itself as a generic procedure for aggregating data with statistically rigorous quality guarantees.

## 1 INTRODUCTION

In many real-world machine learning applications the amount of quality data for training is limited to what is locally available to the data owners. Collaborative learning techniques have shown some potential by allowing multiple participants to contribute data and jointly train a model in a privacy-preserving manner (McMahan et al., 2017), however, relying solely on distributed and diverse private data from collaborators in the learning process significantly affects *personalization* (Blum et al., 2017). Unlike federated learning (McMahan et al., 2017), several practical scenarios consider a data owner having specific learning goals, and not being interested in having a *well-generalized* model that works for the rest of the participants. This indicates that *not all* data, but only quality data, meaning data with *specific attributes*, are relevant for the learning agent, and motivates this work to resolve challenges of quality data acquisition.

We distinguish data agents with distribution mismatch into three categories: (i) honest data agents with unintentionally biased data, (ii) dishonest data agents selling artificial data, and (iii) malicious data agents generating adversarial data (Szegedy et al., 2014; Zhang et al., 2023). For the first case, an example is a hospital interested in training a model to detect a rare disease based on an X-ray image of a patient. Since the disease is rare, the hospital may not have sufficient local data to train an accurate prediction model, but when buying data from other hospitals, or a *data marketplace* (Fernandez, 2023), it is important to ensure similar data conditions, as different X-ray machines and softwares may have been used to take the measurements (Al-qaness et al., 2024). An example of the second case is fake social media accounts sold on a data marketplace as genuine users. For the third case, an example is recommendation systems with fake review injection (Nawara et al., 2025). These examples outline the need for having principled data acquisition/sharing protocols with quality guarantees.

Quality data acquisition approaches have motivated novel data trading or sharing mechanisms, such as *data markets*, where the data agent can prioritize access to relevant data for better personalization (Fernandez et al., 2020). In a nutshell, we assume that *in-distribution* implies relevant data, similar to the training samples, for improving personalization. In contrast, *out-of-distribution* (OOD) indicates outliers and unseen samples, including data with potential contamination, which may hold value for better generalization. A participant's benefit from the data of other participants hinges on careful data selection per its learning goal, as outlined in the earlier real-world examples, leading to the development of efficient data valuation techniques, such as Shapley value (Ghorbani and Zou, 2019) and its variants, amongst others. Unfortunately, these approaches are still limited: first, they are model-specific and computationally expensive; second, they are often used *post-hoc*, i.e., after collecting the data, which might be contaminated and of less value; and third, detecting the optimal set of collaboration partners without making any distributional assumptions on their data is challenging. This raises some fundamental questions in developing data sharing mechanisms prior to training. Furthermore, in a networked system, any of such mechanisms should be compatible with emerging peer-to-peer collaboration protocols between decentralized data sources and AI-powered tools, such as *model context protocol* (MCP) (Hou et al., 2025). From the perspective of MCP, the data agents are the local data sources and an MCP host then wants to access the data from the local data sources which are relevant for personalization. In this premise, the focus of this work is to develop an agile data trading framework between data agents that identifies valuable training data, *without making any distributional assumption*, through the formalization of guarantees to data contamination tests.

**Data contamination model:** Consider that a local data agent observes data $\mathcal{D}_0 = \{(X_i, Y_i)\}_{i=1}^n$ of $n$ input-output pairs, where $X_i \in \mathbb{R}^d$ is the observed feature and $Y_i$ is the response with $Z_i = (X_i, Y_i) \sim P_0$ for an unknown local distribution $P_0$. In the data sharing platform, there are $K$ other data agents with each their own local data $\mathcal{D}_k = \{(X_i^k, Y_i^k)\}_{i=1}^{m_k}$ of $m_k$ input-output pairs with $Z_i^k = (X_i^k, Y_i^k) \sim \tilde{P}_k$ for unknown local distributions $\tilde{P}_k$. Each of the unknown local distributions can be decomposed as $\tilde{P}_k = (1 - \pi_k)P_0 + \pi_k P_k$ where $P_k$ is a proper outlier distribution and $\pi_k \in [0, 1]$ is the contamination factor (Blanchard et al., 2010). If the learning goal is better personalization, as is the focus of our work, finding the best data agent(s) for data acquisition naturally boils down to learning who has the least contaminated data.

**General data sharing procedure and challenges:** *First*, upon the availability of data from $K$ data agents in the data sharing platform, the data requester, say the data agent 0, must decide from which data agent(s) to acquire more data to improve personalization (Lee et al., 2018). Once the first problem of who to pair with is resolved, the data sharing procedure is executed in rounds, where, in each round, a batch of data is acquired and used to decide the data acquisition policy in the next round; the process ends in either of the following cases: (i) when the data has been exhausted, (ii) the data purchasing budget has been depleted, or (iii) the data agent 0 decides that acquiring more data is not necessary (beneficial). *Second*, after the termination of data sharing, data agent 0 uses a data subset selection technique, for instance a complex data valuation technique (Koh and Liang, 2017; Ghorbani and Zou, 2019; Yoon et al., 2020), or a simpler outlier detection method (Schölkopf et al., 2001; Hawkins et al., 2002; Ruff et al., 2018), to filter out OOD datapoints and consequently improve personalization.

Based on the data contamination model, the first challenge can be posed as testing null hypotheses $H_0^1, \ldots, H_0^K$ where $H_0^k : \pi_k \leq \pi_{\text{th}}$ for a user-specified contamination factor threshold $\pi_{\text{th}} \in [0, 1)$. The idea is that we reject $H_0^k$ when we have sufficient evidence that the data from the $k$-th data agent is contaminated beyond the threshold we allow. A theoretical (but impractical) solution to this problem has been considered in Blanchard et al. (2010), and some existing works are occupied with estimating the contamination factor (Ramaswamy et al., 2016; Perini et al., 2023); however, Ramaswamy et al. (2016) provide no distributional guarantees, and Perini et al. (2023) uses a complex Bayesian approach that requires distributional and prior assumptions.

**Main Contributions:** We propose a distribution-free solution for testing $H_0^1, \ldots, H_0^K$ with false discovery rate (FDR) control guarantees that builds on the ideas of conformal outlier detection (Bates et al., 2023; Marandon et al., 2024), and apply the developed tools to the data sharing scenario, as outlined in Figure 1. The summary of our contributions is the following:

- We propose novel two-sample testing procedures by introducing a class of conformal data contamination p-values for testing $H_0^k : \pi_k \leq \pi_{\text{th}}$, $k = 1, \ldots, K$, without any distributional assumptions which provably controls the false rejection probability, and generalize the combination tests of

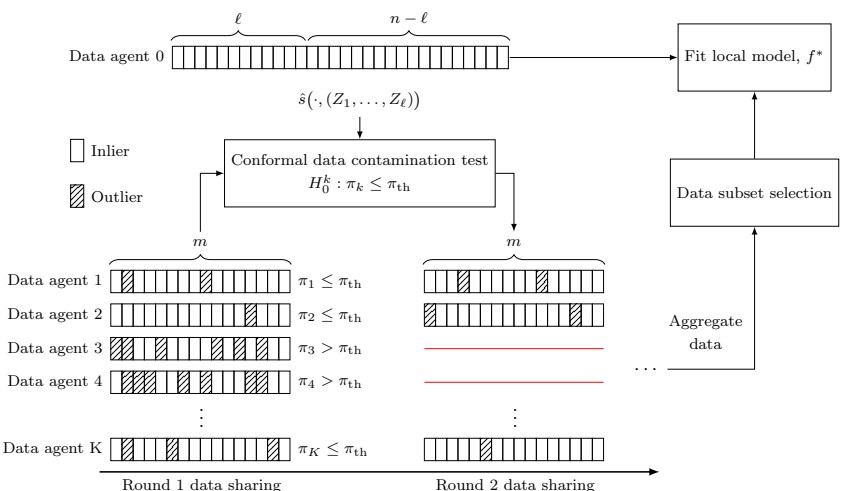

Figure 1: *Proposed data sharing procedure:* In the first round, $m$ samples are received at data agent $0$ from each of the other $K$ data agents, conformal p-values are computed, and the proposed conformal data contamination test is performed. In the following round, data agent $0$ only acquire data from the data agents which was not rejected in the conformal data contamination test. Finally, a data subset selection technique is used to filter away OOD data, and the local model is trained.

Bates et al. (2023) from the special case of $\pi_{\text{th}} = 0$ to any $\pi_{\text{th}} \in [0, 1)$. Moreover, the tests provide a lot of flexibility with the choice of the conformal score, and as such are compatible with a wide range of outlier detection methods.

- We show that the p-value sequence for testing $H_0^1, \ldots, H_0^K$ is positive regression dependent on a subset (PRDS) thereby allowing for FDR control using the Benjamini-Hochberg (BH) procedure.

- We propose a data sharing procedure developed for personalization which uses the conformal data contamination tests to initialize the collaboration of data agents through data sharing while providing theoretical guarantees.

- Numerical experiments are conducted on medical image datasets, as well as a label noise scenario on the MNIST dataset, to validate the effectiveness of the conformal data contamination tests and the proposed data sharing procedure.

**Other related works:** Several works towards collaborative machine learning ecosystems focus on efficient mechanism design that enables data sharing amongst heterogeneous data agents. Based on rich literature from economics and game theory, novel value attribution frameworks have been investigated to answer fundamental questions on data collection and sharing, which precede compute-intensive model training. For instance, Huang et al. (2023) discussed complex interactions between data agents when designing mechanisms for quality and diverse data collection in collaborative learning. Studies on *data marketplaces* started evaluating context and specific goal of data agents to develop data acquisition process, such as in Lu et al. (2024), where a buyer's objective is fulfilled through collection of statistically relevant data for training a predictive model, and in Capitaine et al. (2024), where the total welfare is maximized through collaboration. While *pricing* on data value has been a key incentive for assuring quality data acquisition in most works, Ananthakrishnan et al. (2024) extended it to understand how much test data to use for verification, and how to tune hyperparameters once data collection is done for offering machine learning services.

A line of work in conformal inference, that is related to but different from ours, deals with *corruption* in the calibration data, which in the general sense means that the calibration data is not exactly distributed according to the null distribution $P_0$. Such *corruption* violates the fundamental exchangeability assumption that conformal inference relies on, however, approximately valid inference is still possible. The methodologies in this line of work is relevant for downstream uncertainty quantification following (imperfect) in-distribution data acquisition. Tibshirani et al. (2019) proposed weighted conformal prediction and applied it to handle covariate shift. Subsequently, weighted conformal prediction was considered for label shift in Podkopaev and Ramdas (2021) and for generalized covari-

ate shift and posterior drift in Wang and Qiao (2025). A detailed theoretical analysis of conformal prediction beyond exchangeability was conducted in Barber et al. (2023). Recently *robust* conformal prediction has been considered: Bashari et al. (2025) proposed a technique to avoid *corruption* by detecting outliers in the calibration data, and Feldman and Romano (2024) considered scenarios with access to privileged information (not available at test time) that explains the distribution shift. If further, data agents have strict privacy constraints, federated conformal methodologies can be used for distributed uncertainty quantification. Conformal inference uses the empirical quantile of the calibration scores, and so Humbert et al. (2023) proposed to use the quantile-of-quantiles among data agents, while Lu et al. (2023) proposed a federated quantile estimator. Concurrently, Plassier et al. (2023) developed a technique based on weighted conformal prediction and federated quantile estimation to handle heterogeneous label distributions, which they subsequently adapted to heterogeneity in covariate distributions Plassier et al. (2024). Contrary to this paper, these works deal with uncertainty quantification in the downstream classification task.

## 2 PRELIMINARIES

***Notations :*** Let $[n] = \{1, \ldots, n\}$ and $[n]_0 = \{0, \ldots, n\}$ denote index sets for $n \geq 1$, and in an abuse of notation let $[n]/m = \{1/m, \ldots, n/m\}$.

### 2.1 MULTIPLE TESTING

Consider a multiple testing scenario in which $K$ hypotheses $H_0^1, \ldots, H_0^K$ are tested, and denote by $p_1, \ldots, p_K$ the p-values. We define $S$ as the number of rejected null hypotheses that are actually false, and $V$ as the number of rejected null hypotheses that are actually true (type I error). The total number of rejected hypotheses is given by $R = V + S$, while $K_0$ represents the total number of true null hypotheses.

A widely studied error rate in multiple hypothesis testing is the FDR, introduced by Benjamini and Hochberg (1995). It is defined as $\text{FDR} = \mathbb{E}[V/\max(1, R)]$. Similarly, the power of a test, also known as the true discovery rate (TDR), is defined as $\mathbb{E}[S/\max(1, K - K_0)]$. The most well-known method for controlling the FDR is the BH procedure: at level $q^* \in (0, 1)$, reject $H_0^{(1)}, \ldots, H_0^{(\kappa)}$ with $\kappa = \max\{j \in [K] : p_{(j)} \leq q^* j/K\}$, where $p_{(1)} \leq \cdots \leq p_{(K)}$ are the ordered p-values and $H_0^{(j)}$ is the null hypothesis associated to $p_{(j)}$, hence, the rejection set is $\hat{\mathcal{H}}_1 = \{j \in [K] : p_j \leq p_{(\kappa)}\}$. When the p-values are PRDS (see Section S1.1), the BH procedure controls the FDR at level $q^* K_0/K$ (Benjamini and Yekutieli, 2001). There exists adaptive procedures which estimate $K_0$, defined as $\hat{K}_0$, and accordingly adjusts the procedure by running it at level $q^* = \alpha K/\hat{K}_0$, for significance level $\alpha \in (0, 1)$ (Storey et al., 2003; Benjamini et al., 2006). With Storey's BH procedure, $\kappa_{\alpha,\gamma} = \max\{j \in [K] : p_{(j)} \leq \alpha j/\hat{K}_0\}$, where $\hat{K}_0/K = \sum_{j=1}^K \mathbb{1}[p_j > \gamma]/(1 - \gamma)$ with the hyperparameter $\gamma \in (0, 1)$ controlling the bias-variance trade-off, and we denote the rejection set of this procedure by $\text{SBH}_{\alpha,\gamma}(p_1, \ldots, p_K) = \{j \in [K] : p_j \leq p_{(\kappa_{\alpha,\gamma})}\}$.

A direct approach was introduced in Storey (2002) by considering a liberal estimate of the FDR

$$\widehat{\text{FDR}} = \min\Big\{1, \frac{\delta \sum_{j=1}^K \mathbb{1}[p_j > \gamma]}{(1 - \gamma) \sum_{j=1}^K \mathbb{1}[p_j \leq \delta]}\Big\}, \tag{1}$$

where $\delta \in [0, 1]$ defines the rejection region $[0, \delta]$. Here, one can set $\delta = p_{(\kappa)}$ for a chosen $\kappa$, yielding a rejection set $\hat{\mathcal{H}}_1 = \{j \in [K] : p_j \leq p_{(\kappa)}\}$, and then estimate the corresponding FDR for this $\kappa$.

### 2.2 CONFORMAL OUTLIER DETECTION

In the conformal outlier detection setting, a number of test points $Z_{n+1}, \ldots, Z_{n+m}$, as well as a null sample $Z_1, \ldots, Z_n$ are observed. It is assumed that the null sample datapoints are exchangeable and from an unknown distribution $P_0$. Then, a real-valued conformal score function, $\hat{s}$, is defined satisfying the following permutation invariance property

$$\hat{s}(\cdot, (Z_1, \ldots, Z_\ell), (Z_{\sigma(\ell+1)}, \ldots, Z_{\sigma(n+m)})) = \hat{s}(\cdot, (Z_1, \ldots, Z_\ell), (Z_{\ell+1}, \ldots, Z_{n+m})),$$

for any permutation $\sigma$ of $\{\ell + 1, \ldots, n + m\}$, and $0 \leq \ell < n$ (Marandon et al., 2024). The interpretation here is that a large value of $\hat{s}_{n+i} = \hat{s}(Z_{n+i}, (Z_1, \ldots, Z_\ell), (Z_{\ell+1}, \ldots, Z_{n+m}))$, $i = 1, \ldots, m$, is evidence supporting the hypothesis that $Z_{n+i} \sim P_0$, and vice versa. We will also assume the following property: $(Z_1, \ldots, Z_n, (Z_{n+i})_{i \in \bar{\mathcal{H}}_0})$ are exchangeable conditional on $(Z_{n+i})_{i \in \bar{\mathcal{H}}_1}$, where the set of nulls in the test sample is denoted $\bar{\mathcal{H}}_0 = \{i \in \{1, \ldots, m\} \text{ s.t. } \bar{H}_0^i \text{ is true}\}$ with $\bar{H}_0^i : Z_{n+i} \sim P_0$, and $\bar{\mathcal{H}}_1 = \{1, \ldots, m\} \setminus \bar{\mathcal{H}}_0$. For $i = 1, \ldots, m$, the conformal p-value associated to the test point $Z_{n+i}$ is

$$\hat{p}_i = \frac{1}{n - \ell + 1}\Big(1 + \sum_{j=\ell+1}^{n} \mathbb{1}[\hat{s}_j \leq \hat{s}_{n+i}]\Big). \tag{2}$$

Assuming that the conformal scores are continuously distributed and that $Z_{n+i}$ is an inlier independent of $Z_1, \ldots, Z_\ell$, then $\hat{p}_i$ is uniformly distributed on $[n - \ell + 1]/(n - \ell + 1)$ , and one has that $\lfloor \beta(n - \ell + 1) \rfloor/(n - \ell + 1) \leq \mathbb{P}(\hat{p}_i \leq \beta) \leq \beta$, for $\beta \in (0, 1)$. This shows that $\hat{p}_i$ is a *marginally* valid p-value with almost exact control of the false rejection probability (Marandon et al., 2024).

## 3 MAIN RESULTS

In the first round of data sharing as in Figure 1, the data from the $k$-th agent yields conformal p-values $\hat{p}_1^k, \ldots, \hat{p}_m^k$. In the following, we consider a single sequence of conformal p-values, and so to simplify notations we denote this sequence by $\hat{p}_1, \ldots, \hat{p}_m$. We are interested in testing the null hypothesis $H_0 : \pi \leq \pi_{\text{th}}$ using this sequence of conformal p-values. Such tests can be referred to as combination tests as they use a test statistic which combines a sequence of p-values. We also refer to these tests as two-sample tests since the conformal p-values, and in turn the combination test statistics, are computed using two samples of data, i.e., the null sample and the test sample. We now state one of our main results, defining valid p-values for two natural test statistics.

---

**Theorem 1.** *Let $H_0 : \pi \leq \pi_{\text{th}}$, let $\hat{p}_i$ for $i = 1, \ldots, m$ denote the conformal p-values as introduced in Section 2.2, and denote by $\mathrm{B}_{\pi_{\text{th}}}^m(k)$ the probability mass function of a binomial distribution evaluated at $k$ with parameters $\pi_{\text{th}}$ and $m$. Denote by $F_{\text{NHG}}(x; n + k, n, k - r)$ the cumulative distribution function (CDF) of the negative hypergeometric distribution evaluated at $x \in \{0, 1, \ldots, n\}$ with population size $n + k$, number of success states $n$, and number of failures $k - r$. The following are valid p-values for $H_0$:*

$$\hat{u}^{\text{storey}} = \sum_{k=T^{\text{storey}}+1}^{m} \mathrm{B}_{\pi_{\text{th}}}^m(k) F_{\text{NHG}}\big(\lfloor \lambda(n+1) \rfloor - 1; n + k, n, k - T^{\text{storey}}\big) + \sum_{k=0}^{T^{\text{storey}}} \mathrm{B}_{\pi_{\text{th}}}^m(k), \tag{3}$$

*where $T^{\text{storey}} = \sum_{i=1}^{m} \mathbb{1}[\hat{p}_i > \lambda]$ and $\lambda \in [n]/(n+1)$ is a hyperparameter;*

$$\hat{u}^{\text{quantile}} = \sum_{k=i_0+1}^{m} \mathrm{B}_{\pi_{\text{th}}}^m(k) F_{\text{NHG}}\big(T^{\text{quantile}} - 1; n + k, n, k - i_0\big) + \sum_{k=0}^{i_0} \mathrm{B}_{\pi_{\text{th}}}^m(k), \tag{4}$$

*where $T^{\text{quantile}} = (n+1)\hat{p}_{(m-i_0)}$ with $\hat{p}_{(m-i_0)}$ denoting the $(m-i_0)$-th smallest conformal p-value, and $i_0 \in [m-1]_0$ is a hyperparameter.*

---

*Proof.* The proof is given in Section S1 of the supplementary material. $\square$

**Remark 1.** *The strategy of the proof is to use the law of total probability to decompose the rejection probability under the null into a sum over the number of inliers. Subsequently, an inequality is made by discarding contributions from the conformal p-values of outliers. Finally, using the marginal distribution of the order statistics of the conformal p-values, which specifically is a negative hypergeometric distribution, see Gazin et al. (2024) and Biscio et al. (2025), allows for an explicit expression for the rejection probability under the null, which in turn defines a p-value.*

*The test statistic $T^{\text{storey}}$ is motivated by the Storey estimator of $\pi$ by Storey (2002), while $T^{\text{quantile}}$ is motivated by the quantile estimator of $\pi$ by Benjamini et al. (2006).*

More general test statistics are also allowed, however, constructing valid p-values require knowledge of the distribution of the test statistic under the null. In the following result, we present two other test statistics and their associated p-values which are asymptotically valid.

**Theorem 2.** *Let $H_0 : \pi \leq \pi_{\text{th}}$, let $\hat{p}_i$ for $i = 1, \ldots, m$ denote the conformal p-values as introduced in Section 2.2, and denote by $\mathrm{B}^m_{\pi_{\text{th}}}(k)$ the probability mass function of a binomial distribution evaluated at $k$ with parameters $\pi_{\text{th}}$ and $m$. Then, asymptotically as $n \to \infty$ the following are valid p-values for $H_0$:*

$$\hat{u}^{\text{fisher}} = \pi_{\text{th}}^m + \sum_{k=1}^m \mathrm{B}^m_{\pi_{\text{th}}}(k) F_{\chi^2_{2k}} \left( \frac{-T^{\text{fisher}} - 2k\log\left(\frac{1}{n+1}\right) + 2k(\sqrt{1+k/n} - 1)}{\sqrt{1+k/n}} \right), \quad (5)$$

*where $T^{\text{fisher}} = -2 \sum_{i=1}^m \left( \log\left(\frac{1}{n+1}\right) - \log(\hat{p}_i) \right)$, $F_{\chi^2_{2k}}$ is the CDF of the chi-square distribution with $2k$ degrees of freedom;*

$$\hat{u}^{\text{sum}} = \pi_{\text{th}}^m + \sum_{k=1}^m \mathrm{B}^m_{\pi_{\text{th}}}(k) F_{\mathrm{IH}_k} \left( \frac{T^{\text{sum}} + k(\sqrt{1+k/n} - 1)/2}{\sqrt{1+k/n}} \right), \quad (6)$$

*where $T^{\text{sum}} = \sum_{i=1}^m \hat{p}_i$, $F_{\mathrm{IH}_k}$ is the CDF of the Irwin-Hall distribution with parameter $k$.*

*Proof.* The proof is found in Section S1 of the supplementary material. $\square$

**Remark 2.** *The strategy of the proof is similar as that of Theorem 1, however, since the distribution of $T^{\text{fisher}}$ or $T^{\text{sum}}$ where $\hat{p}_i$ are all inliers is not explicitly known, we employ the asymptotic result Theorem S1 of Bates et al. (2023). The general theorem is actually more general than presented here, as one can consider a general class of test statistics of the form $\sum_{i=1}^m G(\hat{p}_i)$ requiring that $G(U)$ has finite moments for $U \sim \mathrm{Unif}([0,1])$ (Bates et al., 2023). The general statement with precise conditions required on $G$ is given in the supplementary material.*

*The test statistic $T^{\text{fisher}}$ is motivated by Fisher's combination test (Fisher, 1925), and $T^{\text{sum}}$ is another classical combination test statistic (Vovk and Wang, 2020).*

With the p-values of Eqs. (3)-(6), tests for $H_0$ can be conducted without requiring any assumptions on the null distribution, $P_0$, and outlier distribution, $P_1$. Effective conformal data contamination testing relies on conformal scores that clearly separate inliers from outliers, leading to small conformal p-values for outliers and enabling exact control when $\pi = \pi_{\text{th}}$, as seen in the details in the supplementary material. The key parameter is $\ell$, which balances test power: larger $\ell$ improves score quality, while larger $n - \ell$ enhances distribution approximation. Finally, having a large $m$ in general also improves the power of the conformal data contamination tests as more evidence against the null can be aggregated. This is opposite to the case of conformal outlier detection for which Mary and Roquain (2022) studied how larger $m$ results in a loss of power.

In case multiple test sets are observed, we have multiple sets of conformal p-values, each of which gives a conformal data contamination p-value. We consider here how to do multiple testing with this sequence of conformal data contamination p-values while proving FDR control guarantees. Motivated by the result of Bates et al. (2023) that conformal p-values are PRDS, we have shown that the Storey conformal data contamination p-values are indeed also PRDS. Hence, we can simply apply the BH procedure on the conformal data contamination p-values to guarantee FDR control.

**Theorem 3.** *Let $\hat{u}_1^{\text{storey}}, \ldots, \hat{u}_K^{\text{storey}}$ be $K$ Storey conformal data contamination p-values as in Eq. (3), with the $k$-th derived from conformal p-values $\hat{p}_{k,1}, \ldots, \hat{p}_{k,m}$ assuming independence of the $K$ test sets $\mathcal{D}_k = \{(X_i^k, Y_i^k)\}_{i=1}^m$. Then, the Storey conformal data contamination p-values are PRDS.*

*Proof.* The proof is found in Section S1 of the supplementary material. $\square$

**Remark 3.** *Notice that Theorem 3 only refers to the Storey conformal data contamination p-values, and as such we do not provide the same theoretical FDR guarantees when using the other conformal data contamination p-values. However, we expect the result can be generalized to a broad class of conformal data contamination p-values, herein the ones presented in this paper, and we leave this as a line of future investigation.*

**Selecting the test:** Based on theoretical considerations and numerical experiments, we give the following recommendations: For small calibration sets (e.g. $n - \ell < 100$), the Storey and Quantile statistics are generally preferred as they offer finite-sample guarantees; however, both require the selection of a hyperparameter, which can be challenging in practice. We emphasize the parallel to the Storey and Quantile estimators of the true null proportion, see Storey et al. (2003) and Benjamini et al. (2006), of which neither one is consistently better. When ample calibration data are available, the Fisher or Sum statistics are recommended as they are hyperparameter free. The Fisher statistic is suited for low contamination thresholds, whereas the Sum statistic is appropriate for higher contamination thresholds.

## 4  PROPOSED DATA SHARING PROCEDURE

Taking the perspective of the 0-th data agent, we consider a scenario where in the first round each of the $K$ other data agents send some datapoints $Z_i^k$ for $i \in [m]$ and $k \in [K]$ where $m$ is the number of datapoints acquired from each of the other data agents. We assume for simplicity that $m$ is the same for all data agents, however this assumption is not required for the proposed methodology. We model data coming from the $k$-th data agent, $k \in [K]$, to the 0-th data agent by the contamination model (Blanchard et al., 2010): $Z_i^k \sim (1 - \pi_k)P_0 + \pi_k P_k$ where $\pi_k \in [0, 1]$ is the contamination factor.[1] OOD samples $Z_i^k \sim P_k$, also referred to as outliers, are not of interest for the 0-th data agent as we focus on improving personalization. To avoid spending resources on acquiring outliers from other data agents, the 0-th data agent attempts to determine the contamination factors with the purpose to subsequently only collaborate with other data agents having a low contamination factor.

First, conformal p-values, $\hat{p}_i^k$, are computed as in Eq. (2). Using these conformal p-values, we define non-contamination statistics, $T_k$, mapping a sequence of conformal p-values to a positive real number, for which a large value indicates a small contamination factor, and vice versa. The specific options in consideration were discussed in Section 3 where we also presented p-values, denoted $\hat{u}_k$.

Given a collaboration budget for subsequent rounds, meaning that going forward we can at most acquire data from $K_{\text{budget}}$ other data agents, we decide to collaborate with the $K_{\text{budget}}$ data agents with largest non-contamination statistic, and denote this index set as $\hat{\mathcal{H}}_0 = \{\sigma(i) : i \in [K_{\text{budget}}]\}$ where $\sigma$ is a permutation on $[K]$ such that $T_{\sigma(1)} \geq T_{\sigma(2)} \geq \cdots \geq T_{\sigma(K)}$. Using the conformal data contamination p-values, $\hat{u}_k$, we may estimate the FDR for null hypotheses $H_0^k : \pi_k \leq \pi_{\text{th}}$, $\pi_{\text{th}} \in [0, 1)$, e.g., using Storey's direct approach Eq. (1), to gain insights into the decision. If we are not given a collaboration budget, but instead a threshold on how much contamination we tolerate, i.e., $\pi_{\text{th}}$, we may consider testing null hypotheses $H_0^k : \pi_k \leq \pi_{\text{th}}$. For a specified significance level, $\alpha \in (0, 1)$, this can be done with the conformal data contamination p-values using an adaptive BH procedure, e.g., Storey's BH procedure described in Section 2.1, yielding a set of non-rejected data agents, $\hat{\mathcal{H}}_0 = [K] \setminus \text{SBH}_{\alpha,\gamma}(\hat{u}_1, \ldots, \hat{u}_K)$ for Storey's hyperparameter $\gamma \in (0, 1)$.

After determining which data agents to collaborate with in the following round(s), the 0-th data agent once again acquires data $Z_{m+i}^k$ for $i \in [m]$ and $k \in \hat{\mathcal{H}}_0$, and can then aggregate all the received data, followed by data subset selection for instance using techniques from data valuation (Koh and Liang, 2017; Ghorbani and Zou, 2019; Yoon et al., 2020). The selected data is combined with the $n$ local datapoints, and the 0-th data agent trains its local model. The proposed procedure is visualized in Figure 1 and also summarized in Section S2 of the supplementary material. An overview of the scenario variables and hyperparameters is given in Section S2.1, and in Section S4.5 a data-driven approach to hyperparameter selection is outlined.

---

[1]We make the assumption here that $P_k$ are proper novelty distributions with respect to $P_0$: there exists no decomposition of the form $P_k = (1 - \zeta)Q + \zeta P_0$ where $Q$ is a probability distribution and $0 < \zeta \leq 1$, see Blanchard et al. (2010); Zhu et al. (2023) for details.

**Complexity:** One of the strengths of the proposed data acquisition protocol is its low computational complexity due to preceding the expensive model training procedure. Note that computing a conformal data contamination p-value as defined in Eqs. (3)-(6) requires at most $m$ evaluations of known CDFs: for Quantile and Storey the CDF of a negative hypergeometric distribution, for Fisher the CDF of a chi-square distribution, and for Sum the CDF of an Irwin-Hall distribution. The most computationally complex part of the data acquisition protocol is fitting of the conformal score function $\hat{s}$ but notably the methodology complies with a wide range of conformal scores with the choice open to the practitioner. Further, the procedure scales well to a large number of data agents by virtue of the Benjamini-Hochberg procedure which was designed for large scale testing.

**Limitations:** The proposed data acquisition procedure has some limitations. First, the novel conformal data contamination tests require exchangeability of calibration and test data to be theoretically valid, restricting use in settings like time series. Second, we made the deliberate choice to focus on the novel conformal data contamination tests, rather than aspects of data subset selection, however, in a practical procedure we recommend to include some data subset selection (Ghorbani and Zou, 2019), or data weighting (Ding and Wang, 2022). Lastly, the proposed data sharing procedure is not privacy-aware, limiting applicability to scenarios with strict privacy constraints.

## 5 NUMERICAL EXPERIMENTS AND ANALYSIS

In this section, we numerically evaluate the proposed methodology on two data contamination scenarios. In the first scenario, retinal fundus images are used to classify severity of diabetic retinopathy and the inliers are drawn from RetinaMNIST (Liu et al., 2022) with contaminated data drawn from EyePACS (Dugas et al., 2015; Gulshan et al., 2016). In the second scenario, images of handwritten digits are used to classify the digits (1, 4, and 7) and the inliers are drawn noise-free from MNIST (Lecun et al., 1998) with contaminated data having incorrect labels. First we introduce the data setup, and present the baseline approaches. Then, the conformal data contamination tests are analyzed numerically. Finally, the performance of the proposed data sharing procedure is evaluated on the classification tasks.

### 5.1 DATA SETUP AND BASELINES

The RetinaMNIST and EyePACS datasets consists of 1600 and 35126 retinal fundus images, respectively. Each image in RetinaMNIST is $28 \times 28 \times 3$ with each pixel taking values in $[255]_0$ and has an associated label $y \in [4]_0$ indicating the severity of diabetic retinopathy ranging from no diabetic retinopathy to proliferate retinopathy. The EyePACS data has varying resolutions, so we crop and downsample the images to the $28 \times 28 \times 3$ resolution.

The MNIST dataset restricted to digits 1, 4, and 7 consists of 21994 images of handwritten digits. Each image is $28 \times 28$ with each pixel taking values in $[255]_0$ and has an associated label $y \in [2]_0$ indicating the digit.

We normalize the images to the interval $[0, 1]$ by dividing pixel-wise with 255. In all cases we sample randomly $n$ in-distribution datapoints for the 0-th data agent. Meanwhile, the $k$-th data agent observes $2m$ datapoints and the number of OOD datapoints is sampled from $\mathrm{B}_{\pi_k}^{2m}$. For all the numerical experiments we set $n = 100$, $\ell = 60$, $m = 40$, $K = 10$, $\lambda = \lfloor n/8 \rfloor/(n+1)$, $i_0 = \lfloor m/3 \rfloor$, and $\gamma = 0.5$. Unless otherwise specified, we simulate the data contamination factors independent and identically distributed (iid) according to $\pi_k \sim \mathrm{Uniform}([0, 1])$.

As a conformal score, we will use the *positive unlabelled* (PU) classification approach advocated in Marandon et al. (2024). For simplicity, the PU classifier is chosen as logistic regression (LR), and we use the standard implementation in *scikit-learn* with balanced class weights. We define the score to be class-wise in the sense that

$$\hat{s}((\boldsymbol{X}, Y), (Z_1, \ldots, Z_\ell), (Z_{\ell+1}, \ldots, Z_{n+m})) = \sum_{i \in \mathcal{Y}} \mathbb{1}[Y = i] \, \hat{s}_i(\boldsymbol{X}, \boldsymbol{X}_{1:\ell}^{Y=i}, \boldsymbol{X}_{\ell+1:n+m}^{Y=i}),$$

where $\mathcal{Y}$ denotes the set of labels, $\boldsymbol{X}_{1:\ell}^{Y=i} = (\boldsymbol{X}_j : Y_j = i, j \in \{1, \ldots, \ell\})$ denotes the subset of $\boldsymbol{X}_1, \ldots, \boldsymbol{X}_\ell$ for which $Y_j = i$ for $j \in \{1, \ldots, \ell\}$. Specifically, $\hat{s}_i(\boldsymbol{X}, \boldsymbol{X}_{1:\ell}^{Y=i}, \boldsymbol{X}_{\ell+1:n+m}^{Y=i})$ denotes the predicted probability that $(\boldsymbol{X}, i)$ has label 1, i.e., is an inlier, using LR fitted to $\boldsymbol{X}_{1:\ell}^{Y=i}$ with labels 1 and $\boldsymbol{X}_{\ell+1:n+m}^{Y=i}$ with labels $-1$.

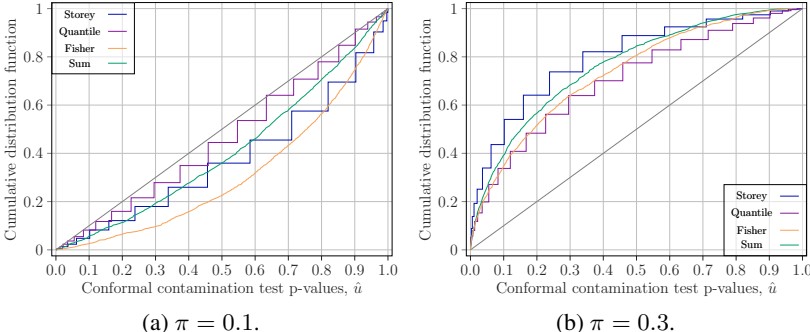

Figure 2: Empirical CDF of conformal data contamination p-values with $\pi_{\mathrm{th}} = 0.1$ for (a) $\pi = 0.1$ and (b) $\pi = 0.3$. The gray line is the CDF of a standard uniform.

Table 1: (a) TDR and (b) FDR computed from 2000 simulations for varying data contamination thresholds, $\pi_{\mathrm{th}}$, at significance level $\alpha = 0.1$ using Storey's BH procedure with parameter $\gamma = 0.5$.

(a) Data contamination factors are simulated as $\pi_k \sim \mathrm{Uniform}([0, 1])$.

| | $\pi_{\mathrm{th}} = 0$ | $\pi_{\mathrm{th}} = 0.1$ | $\pi_{\mathrm{th}} = 0.2$ |
|---|---|---|---|
| Storey | 0.9128 | 0.8147 | 0.7033 |
| Quantile | 0.8104 | 0.7851 | 0.7537 |
| Fisher | 0.8890 | 0.7576 | 0.6077 |
| Sum | 0.8778 | 0.7994 | 0.7142 |

(b) For $K/2$ of the data agents $\pi_k = \pi_{\mathrm{th}}$ while for the others $\pi_k = 1 - \pi_{\mathrm{th}}$.

| | $\pi_{\mathrm{th}} = 0$ | $\pi_{\mathrm{th}} = 0.1$ | $\pi_{\mathrm{th}} = 0.2$ |
|---|---|---|---|
| Storey | 0.1064 | 0.0708 | 0.0489 |
| Quantile | 0.1120 | 0.1074 | 0.1050 |
| Fisher | 0.0641 | 0.0281 | 0.0153 |
| Sum | 0.0890 | 0.0658 | 0.0458 |

As a classifier we consider support vector classification (SVC) (Boser et al., 1992; Cortes and Vapnik, 1995) for simplicity and robustness, again using the standard implementation in *scikit-learn* (Pedregosa et al., 2011).

We consider the following baselines: (i) *Oracle 1:* The complete oracle knows the true contamination factors and exactly which datapoints are contaminated. Accordingly, it selects for collaboration in the second round the data agents with data contamination factor $\pi_k$ below a specified threshold $\pi_{\mathrm{th}}$, and then uses only the in-distribution data for subsequent model training. (ii) *Oracle 2:* The partial oracle knows the true contamination factors but not which datapoints are contaminated. Accordingly, it selects for collaboration in the second round the data agents with data contamination factor $\pi_k$ below a specified threshold $\pi_{\mathrm{th}}$, and then uses all collected data for subsequent model training. (iii) *Random:* The naïve baseline randomly selects $K_{\mathrm{budget}}$ agents to collaborate with in the second round, and then uses all collected data for subsequent model training.

In Section S3 of the supplementary material we provide an ablation study with synthetic Gaussian data. We also provide additional numerical experiments for the data contamination scenario presented here, as well as six other data contamination scenarios, in Section S4, herein, also validating scalability to larger dataset sizes in Section S4.4.

## 5.2 CONFORMAL DATA CONTAMINATION TESTS

In this section the conformal data contamination tests are numerically analyzed on the retinal fundus images. We showcase in Figure 2 the (empirical) CDF of the conformal data contamination p-values with $\pi_{\mathrm{th}} = 0.1$ and for $\pi = 0.1, 0.3$. From Figure 2a we observe that the conformal data contamination tests are superuniform under the null, as dictated by the theory in Section 3, and notice that the Quantile p-value is nearly standard uniform, meanwhile the Fisher p-value is the most conservative in this case. Figure 2b shows that in this case the Storey test is the most powerful followed by the Sum and Fisher tests, and finally the weakest is the Quantile test.

We report in Table 1a the (empirical) TDR for the conformal data contamination with varying data contamination thresholds $\pi_{\mathrm{th}} \in \{0, 0.1, 0.2\}$ at significance level $\alpha = 0.1$ when using Storey's BH procedure with Storey's hyperparameter $\gamma = 0.5$. We observe the highest TDR with the Storey test and a generally low TDR with the Quantile test, and notably, the Storey and Fisher tests perform well

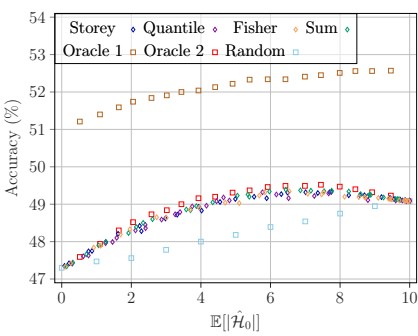 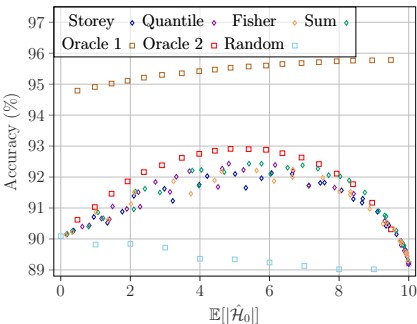

(a) Retinal fundus images. Accuracy without data sharing: 47.77 %.

(b) Handwritten digits. Accuracy without data sharing: 90.75 %.

Figure 3: Plot of the classification accuracy against the average number of collaborating data agents.

at low $\pi_{\mathrm{th}}$ but tend to decrease rapidly in TDR as the data contamination threshold increases, while the decrease is less pronounced with the Quantile and Sum tests.

In Table 1b we numerically validate the FDR control provided by the conformal data contamination tests on the boundary of the null hypothesis. Here, $K_0 = K/2$ data agents have data contamination factor $\pi_k = \pi_{\mathrm{th}}$, and the remaining data agents have $\pi_k = 1 - \pi_{\mathrm{th}}$ so approximately half of the test data is contaminated. Through this construction, we are on the boundary of the null hypothesis, i.e., there are no $\pi_k < \pi_{\mathrm{th}}$ contributing to excess conservativeness from being in the interior of the null hypothesis. Once again, we used Storey's BH procedure at significance level $\alpha = 0.1$ and with Storey's hyperparameter $\gamma = 0.5$. The results show that the tests effectively controls the FDR at the nominal level, and we notice that the Quantile test is the least conservative among the four tests.

### 5.3 COLLABORATIVE DATA SHARING

We show in Figure 3 the relation between the average number of collaborating data agents, $\mathbb{E}[|\hat{\mathcal{H}}_0|]$, and the average classification accuracy across 500 data simulations for both data contamination scenarios, with various hyperparameter settings $(\pi_{\mathrm{th}}, \alpha) \in \{0, 0.1, 0.2, 0.3, 0.4, 0.5, 0.6, 0.7\} \times \{0.05, 0.2, 0.5, 0.7\}$. The proposed data sharing procedures outperform no data sharing and the *random* baseline, even reaching the performance of *oracle 2*, showing the efficiency of the proposed methodology, and highlighting the accuracy improvements achieveable by carefully selecting data agents for collaboration. Further, the accuracy gap between *oracle 1* and *oracle 2* indicates that additional accuracy improvements can be achieved by using data subset selection techniques, although this has not been considered in this work, as our focus is the proposed conformal data contamination tests.

The biggest gains in accuracy can be observed in Figure 3b showing the case of label noise on the MNIST data. Specifically, at $\mathbb{E}[|\hat{\mathcal{H}}_0|] \approx 6$ we achieve an increase in accuracy of up to 3 percentage points with the proposed methods compared to the *random* baseline. For the other scenario, at $\mathbb{E}[|\hat{\mathcal{H}}_0|] \approx 6$ we achieve an increase in accuracy just below 1 percentage point.

## 6 CONCLUSION

We have presented a data sharing framework which selects collaboration partners using novel distribution-free testing procedures, named *conformal data contamination tests*. Leveraging conformal p-values, our method detects agents whose data exceed a contamination threshold while providing false discovery rate guarantees. This enables strategic acquisition of high-quality and personalized data while avoiding irrelevant or harmful data sources. Experiments validate the effectiveness and practicality of our approach.

Our framework opens up several promising directions for future work, including integration with incentive mechanisms in real-world data markets, and extension to continual learning with online data acquisition policies.

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

# SUPPLEMENTARY MATERIAL FOR "CONFORMAL DATA CONTAMINATION TESTS FOR IN-DISTRIBUTION DATA ACQUISITION"

**Anonymous authors**

## S1 CONFORMAL DATA CONTAMINATION TESTS: THEORETICAL DETAILS AND PROOFS

We consider the scenario where we have observed a null sample $\mathcal{D}_{\text{null}} = \{Z_i\}_{i=1}^n$, $Z_i \sim P_0$, as well as a test sample $\mathcal{D}_{\text{test}} = \{Z_{n+j}\}_{j=1}^m$, $Z_{n+j} \sim P_j$. We call data from $P_0$ inliers and data from some other distribution $P_1$ outliers. We define $\bar{\mathcal{H}}_0 = \{j \in [m] : Z_{n+j} \sim P_0\}$, and $\bar{\mathcal{H}}_1 = [m] \setminus \bar{\mathcal{H}}_0$. Following the conformal outlier detection procedure as described in Section 2.2, conformal p-values are computed $\hat{p}_1, \ldots, \hat{p}_m$. As in Section 2.2 we shall assume that the conformal score is continuously distributed, or almost surely has no ties, and that the inliers are exchangeable conditioned on the outliers (Bates et al., 2023).

Two classical estimators of the contamination factor $\pi$ are the Storey estimator of Storey et al. (2003) and the quantile estimator of Benjamini et al. (2006) defined respectively as

$$\hat{\pi}^{\text{storey}} = 1 - \frac{\sum_{i=1}^m \mathbb{1}[\hat{p}_i > \lambda]}{m(1-\lambda)},$$

$$\hat{\pi}^{\text{quantile}} = 1 - \frac{i_0 + 1}{m(1 - \hat{p}_{(m-i_0)})},$$

where $\hat{p}_{(1)} \leq \cdots \leq \hat{p}_{(m)}$ are the ordered conformal p-values. Here $\lambda \in [n]/(n+1)$ and $i_0 \in [m-1]_0$ are hyperparameters controlling the bias-variance trade-off. For these two estimators, the parts related to the data are $\sum_{i=1}^m \mathbb{1}[\hat{p}_i > \lambda]$ and $\hat{p}_{(m-i_0)}$, respectively, and so in the following we will use these as test statistics, noting that small values of these tests statistics will yield large estimates of $\pi$ and so small values are evidence against the null hypothesis $H_0 : \pi \leq \pi_{\text{th}}$.

In the following theorem, we consider the probability of rejecting under $H_0$ when using $T^{\text{storey}} = \sum_{i=1}^m \mathbb{1}[\hat{p}_i > \lambda]$ as the test statistic. Bounding this probability will reveal how to construct a valid p-value for $H_0$.

---

**Theorem S1.** *Let $T^{\text{storey}} = \sum_{i=1}^m \mathbb{1}[\hat{p}_i > \lambda]$ be the test statistic parametrised by $\lambda \in [n]/(n+1)$ for conformal p-values $\hat{p}_1, \ldots, \hat{p}_m$, and consider a rejection region given as $\{0, \ldots, r\}$ for $r \in \{0, 1, \ldots, m\}$. Then, the probability of rejection under the null hypothesis $H_0 : \pi \leq \pi_{\text{th}}$, $\pi_{\text{th}} \in [0, 1)$, is upper bounded by*

$$\mathbb{P}_{H_0}(T^{\text{storey}} \leq r) \leq \sum_{k=r+1}^m \mathrm{B}_{\pi_{\text{th}}}^m(k) F_{\text{NHG}}\big(\lfloor \lambda(n+1) \rfloor - 1; n+k, n, k-r\big) + \sum_{k=0}^r \mathrm{B}_{\pi_{\text{th}}}^m(k), \quad (7)$$

*where $F_{\text{NHG}}(x; n+k, n, k-r)$ is the CDF of the negative hypergeometric distribution evaluated at $x \in \{0, 1, \ldots, n\}$ with population size $n+k$, number of success states $n$, and number of failures $k-r$, and $\mathrm{B}_{\pi_{\text{th}}}^m(k)$ is the probability mass function of a Binomial distribution evaluated at $k$ with parameters $\pi_{\text{th}}$ and $m$. Moreover, if $\mathbb{P}(\hat{p}_i \leq \lambda, \forall i \in \bar{\mathcal{H}}_1) = 1$ and $\pi = \pi_{\text{th}}$ the inequality is exact.*

---

*Proof.* By the law of total probability

$$\mathbb{P}_{H_0}(T^{\text{storey}} \le r) = \sum_{k=0}^{m} \mathbb{P}_{H_0}(k)\mathbb{P}\Big( \sum_{i=1}^{k} \mathbb{1}[\hat{p}_i > \lambda] + \sum_{i=k+1}^{m} \mathbb{1}[\hat{p}_i > \lambda] \le r \mid \bar{\mathcal{H}}_0 = [k]\Big)$$

$$\le \sup_{\pi_0 \in [0, \pi_{\text{th}}]} \sum_{k=0}^{m} \mathrm{B}_{\pi_0}^m(k)\mathbb{P}\Big( \sum_{i=1}^{k} \mathbb{1}[\hat{p}_i > \lambda] \le r \mid \bar{\mathcal{H}}_0 = [k]\Big). \qquad (8)$$

The inequality is tight when $\hat{p}_i \le \lambda$ for all $i \in \mathcal{H}_1$ which with a well chosen $\lambda$ can occur if the conformal score separates well the inliers from the outliers. The distribution of the null data points is binomial with success probability $\pi_0 \in [0, \pi_{\text{th}}]$,

$$\mathrm{B}_{\pi_0}^m(k) = \binom{m}{k}(1-\pi_0)^k \pi_0^{m-k}.$$

The probability of the sum can be expressed in terms of the marginal CDF of the ordered conformal p-values as

$$\mathbb{P}\Big( \sum_{i=1}^{k} \mathbb{1}[\hat{p}_i > \lambda] \le r \mid \bar{\mathcal{H}}_0 = [k]\Big) = \begin{cases} 1 & \text{if } k \le r, \\ \mathbb{P}\big(\hat{p}_{(k-r)} \le \lambda\big) & \text{otherwise,} \end{cases}$$

where $\hat{p}_{(1)} \le \hat{p}_{(2)} \le \cdots \le \hat{p}_{(k)}$ are the ordered conformal p-values among the $k$ p-values from the null. Notably, $\hat{p}_{(k-r)}$ is distributed according to a negative hypergeometric distribution with population size $n + k$, number of success states $n$, and number of failures $k - r$ supported on $\{1, 2, \dots, n+1\}$, see Corollary A.1 of Biscio et al. (2025), and the CDF can be expressed as

$$\mathbb{P}\big(\hat{p}_{(k-r)} \le \lambda\big) = F_{\text{NHG}}\big(\lfloor \lambda(n+1) \rfloor - 1; n+k, n, k-r\big)$$
$$= \frac{n!k!}{(n+k)!}\frac{(x+k-r-1)!(n-x+r)!}{x!(n-x)!(k-r-1)!r!} {}_3F_2\!\left[\begin{matrix} 1, -x, n-x+r+1 \\ n-x+1, 1-x-k+r \end{matrix}; 1\right],$$

where $x := \lfloor \lambda(n+1) \rfloor - 1$, and ${}_3F_2$ is the generalized hypergeometric function.

To conclude the proof, observe that the supremum in Eq. (8) occurs at $\pi_{\text{th}}$, since $\sum_{k=0}^{l} \mathrm{B}_{\pi_0}^m(k)$, $l \in \{0, \dots, m\}$ is a non-decreasing function in $\pi_0$, and $\mathbb{P}(\sum_{i=1}^{k} \mathbb{1}[\hat{p}_i > \lambda] \le r \mid \bar{\mathcal{H}}_0 = [k])$ is a decreasing function of $k$ as the indicator function is non-negative. $\qquad \square$

We have now specified everything necessary to compute an upper bound on the rejection probability (under the null). From this it follows that the corresponding p-value is given by

$$\hat{u}^{\text{storey}} = \sum_{k=T^{\text{storey}}+1}^{m} \mathrm{B}_{\pi_{\text{th}}}^m(k)F_{\text{NHG}}\big(\lfloor \lambda(n+1) \rfloor - 1; n+k, n, k-T^{\text{storey}}\big) + \sum_{k=0}^{T^{\text{storey}}} \mathrm{B}_{\pi_{\text{th}}}^m(k). \quad (9)$$

By construction this p-value is *marginally* valid, i.e., $\mathbb{P}_{H_0}(\hat{u}^{\text{storey}} \le \alpha) \le \alpha$, and the p-value is discretely distributed with at most $m + 1$ levels (including 0 and 1), since $T^{\text{storey}}$ is discretely distributed on $\{0, 1, \dots, m\}$. Hence, not all significance levels $\alpha \in (0, 1)$ can be reached, however, as $m$ grows we can get arbitrarily close.

As a corollary, we present the corresponding upper bound on the rejection probability under $H_0$ when using $(n+1)\hat{p}_{(m-i_0)}$ as our test statistic.

**Corollary S1.** *Let $T^{\text{quantile}} = (n+1)\hat{p}_{(m-i_0)}$ be the test statistic parametrised by $i_0 \in [m-1]_0$ for conformal p-values $\hat{p}_1, \ldots, \hat{p}_m$, and consider a rejection region given as $\{1, \ldots, r\}$ for $r \in \{1, 2, \ldots, n+1\}$. Then, the probability of rejection under the null hypothesis $H_0 : \pi \leq \pi_{\text{th}}$, $\pi_{\text{th}} \in [0, 1)$, is upper bounded by*

$$\mathbb{P}_{H_0}(T^{\text{quantile}} \leq r) \leq \sum_{k=i_0+1}^{m} \mathrm{B}_{\pi_{\text{th}}}^m(k) F_{\text{NHG}}\big(r-1; n+k, n, k-i_0\big) + \sum_{k=0}^{i_0} \mathrm{B}_{\pi_{\text{th}}}^m(k). \quad (10)$$

*Moreover, if $\mathbb{P}(\hat{p}_i \leq r/(n+1), \forall i \in \bar{\mathcal{H}}_1) = 1$ and $\pi = \pi_{\text{th}}$ the inequality is exact.*

*Proof.* The proof follows immediately by noticing that

$$\mathbb{P}\Big(\hat{p}_{(m-i_0)} \leq \frac{r}{n+1}\Big) = \mathbb{P}\Big(\sum_{i=1}^{m} \mathbb{1}\big[\hat{p}_i > \frac{r}{n+1}\big] \leq i_0\Big),$$

and then using Theorem S1. $\qquad \square$

The p-value is given by

$$\hat{u}^{\text{quantile}} = \sum_{k=i_0+1}^{m} \mathrm{B}_{\pi_{\text{th}}}^m(k) F_{\text{NHG}}\big(T^{\text{quantile}} - 1; n+k, n, k-i_0\big) + \sum_{k=0}^{i_0} \mathrm{B}_{\pi_{\text{th}}}^m(k). \quad (11)$$

This is a valid p-value and takes up to $n+1$ unique values. Comparing the p-values proposed in the preceding, a type of duality is observed, in which the roles of the test statistics and the hyperparameters are swapped between the two.

**Remark S1.** *A possible generalization would be to consider a rejection region on a vector of the ordered p-values, for instance considering two of the ordered p-values rather than just one. In such a case, deriving the upper bound would require evaluating the pairwise distribution of the ordered conformal p-values, and the p-value would depend on two hyperparameters rather than one.*

Bates et al. (2023) showed a general result regarding the asymptotic distribution of test statistics of the form $\sum_{i=1}^{m} G(p_i)$, for some general class of functions $G$. In their work, this was used to formulate a correction to the Fisher combination test yielding a valid testing procedure using conformal p-values for the special case of $\pi_{\text{th}} = 0$. In the following theorem, we generalize their result to the setting of this paper, thereby paving the way for constructing more general test statistics which are valid asymptotically.

**Theorem S2.** *Let $T_G = \sum_{i=1}^{m} G(\hat{p}_i)$ be a test statistic for conformal p-values $\hat{p}_1, \ldots, \hat{p}_m$ and an increasing function $G : [0, 1] \to [0, \infty)$ satisfying*

*(i) $\int_0^1 G^{2+\eta}(u)du < \infty$;*

*(ii) $|\frac{1}{n+1} \sum_{j=1}^{n+1} G^k(j/(n+1)) - \int_0^1 G^k(u)du| = o(1/\sqrt{n})$, for $k \in \{1, 2\}$;*

*(iii) $\max_{j \in \{1, \ldots, n+1\}} G(j/(n+1)) = o(\sqrt{n})$.*

*Then, under the null hypothesis $H_0 : \pi \leq \pi_{\text{th}}$, $\pi_{\text{th}} \in [0, 1)$, if $m = \lfloor \gamma n \rfloor$ for some $\gamma > 0$, as $n \to \infty$*

$$\mathbb{P}_{H_0}(T_G \leq r) \leq \pi_{\text{th}}^m + \sum_{k=1}^{m} \mathrm{B}_{\pi_{\text{th}}}^m(k) F_{G^k}\left(\frac{r + k(\sqrt{1+\gamma_k} - 1)\int_0^1 G(u)\mathrm{d}u}{\sqrt{1+\gamma_k}}\right), \quad (12)$$

*where $F_{G^k}$ is the CDF of $\sum_{i=1}^{k} G(U_i)$, $U_i \overset{iid.}{\sim} \text{Unif}([0,1])$, and $\gamma_k = k/n$. Moreover, if $\mathbb{P}(G(\hat{p}_i) = 0, \forall i \in \bar{\mathcal{H}}_1) = 1$ and $\pi = \pi_{\text{th}}$ the inequality is exact.*

*Proof.* By the law of total probability

$$\mathbb{P}_{H_0}\Big(\sum_{i=1}^m G(\hat{p}_i) \le r\Big) = \sum_{k=0}^m \mathbb{P}_{H_0}(k)\mathbb{P}\Big(\sum_{i=1}^k G(\hat{p}_i) + \sum_{i=k+1}^m G(\hat{p}_i) \le r \mid \bar{\mathcal{H}}_0 = [k]\Big)$$

$$\le \sup_{\pi_0 \in [0,\pi_{\text{th}}]} \sum_{k=0}^m \mathrm{B}_{\pi_0}^m(k)\mathbb{P}\Big(\sum_{i=1}^k G(\hat{p}_i) \le r \mid \bar{\mathcal{H}}_0 = [k]\Big), \qquad (13)$$

Now, by Theorem S1 of Bates et al. (2023), when $k = \lfloor \gamma_k n \rfloor$ we have that as $n \to \infty$

$$\mathbb{P}\Big(\sum_{i=1}^k G(\hat{p}_i) \le r \mid \bar{\mathcal{H}}_0 = [k]\Big) \to q_k^*,$$

where $q_k^*$ is such that $r = \sqrt{1 + \gamma_k}Q_{G^k}(q_k^*) - k(\sqrt{1 + \gamma_k} - 1)\int_0^1 G(u)\mathrm{d}u$, and $Q_{G^k}(q_k^*)$ is the $q_k^*$ quantile of $\sum_{i=1}^k G(U_i)$. Isolating $q_k^*$ yields

$$q_k^* = F_{G^k}\left(\frac{r + k(\sqrt{1 + \gamma_k} - 1)\int_0^1 G(u)\mathrm{d}u}{\sqrt{1 + \gamma_k}}\right).$$

It follows that

$$\lim_{n\to\infty} \mathbb{P}_{H_0}\Big(\sum_{i=1}^m G(\hat{p}_i) \le r\Big) \le \pi_{\text{th}}^m + \sum_{k=1}^m \mathrm{B}_{\pi_{\text{th}}}^m(k)F_{G^k}\left(\frac{r + k(\sqrt{1 + \gamma_k} - 1)\int_0^1 G(u)\mathrm{d}u}{\sqrt{1 + \gamma_k}}\right),$$

where we use that $q_k^*$ is a decreasing sequence. □

It follows from the theorem that an asymptotically valid p-value is

$$\hat{u}_G = \pi_{\text{th}}^m + \sum_{k=1}^m \mathrm{B}_{\pi_{\text{th}}}^m(k)F_{G^k}\left(\frac{T_G + k(\sqrt{1 + \gamma_k} - 1)\int_0^1 G(u)\mathrm{d}u}{\sqrt{1 + \gamma_k}}\right). \qquad (14)$$

**Remark S2.** *We have altered the formulation slightly compared to Bates et al. (2023) since we want the function $G$ to map into non-negative numbers and being an increasing function such that a small value of the test statistic is evidence against the null, thereby allowing the inequality of Eq. (13). Moreover, we want $G$ to be such that $\mathbb{P}(G(\hat{p}_i) \le x, \forall i \in \bar{\mathcal{H}}_1)$ is large for small $x$, which is an important property to have a tight approximation in Eq. (13).*

**Remark S3.** *A variant of the Fisher combination test uses $G(\hat{p}) = -2\log\big(\frac{1}{n+1}\big) + 2\log(\hat{p})$, thereby maintaining the shape of the typical Fisher combination function, while making it an increasing function. The result of Bates et al. (2023) can be exploited in this case by noticing that*

$$\mathbb{P}\Big(-2k\log\Big(\frac{1}{n+1}\Big) + 2\sum_{i=1}^k \log(\hat{p}_i) \le r\Big) = 1 - \mathbb{P}\Big(-2\sum_{i=1}^k \log(\hat{p}_i) \le -2k\log\Big(\frac{1}{n+1}\Big) - r\Big).$$

*Particularly, it holds that using this test statistic is equivalent to using the typical Fisher test statistic when $\pi_{\text{th}} = 0$. The technical conditions (i)-(iii) regarding the function $G$ can be verified, see Remark S1 of Bates et al. (2023) for details.*

**Remark S4.** *A Sum combination test uses $G(\hat{p}) = \hat{p}$ for which $\int_0^1 G(u)\mathrm{d}u = 1/2$ and $G(U) \overset{d}{=} \mathrm{Unif}([0,1])$, resulting in $\sum_{i=1}^k G(U_i) \overset{d}{=} \mathrm{IH}(k)$ where $U_i \overset{iid.}{\sim} \mathrm{Unif}([0,1])$ and $\mathrm{IH}(k)$ is the Irwin-Hall distribution with parameter $k$. The technical conditions (i)-(iii) regarding the function $G$ can be verified: condition (i) trivially holds since a polynomial of finite order is finite; for condition (ii) we note that $G(u)$ is increasing and $G'(u) = 1$, thus for $k \in \{1, 2\}$*

$$\left|\frac{1}{n+1}\sum_{j=1}^{n+1}\frac{j^k}{(n+1)^k} - \int_0^1 u^k \mathrm{d}u\right| \le \sum_{j=1}^{n+1}\left|\frac{1}{n+1}\frac{j^k}{(n+1)^k} - \int_{(j-1)/(n+1)}^{j/(n+1)} u^k \mathrm{d}u\right|$$

$$\le \sum_{j=1}^{n+1}\int_{(j-1)/(n+1)}^{j/(n+1)}\left|\frac{j^k}{(n+1)^{k+1}} - u^k\right|\mathrm{d}u$$

$$\le \sum_{j=1}^{n+1}\frac{kj^{k-1}}{(n+1)^{k+1}} = \mathrm{O}\Big(\frac{1}{n}\Big)$$

*where the first two inequalities are due to the triangle inequality, and the third inequality follows by the mean value theorem and the chain rule; finally condition (iii) immediately holds as $\max_{j \in \{1,\dots,n+1\}} G(j/(n+1)) = 1$.*

### S1.1 MULTIPLE TESTING WITH COMBINATION TEST P-VALUES

For completeness of presentation we begin by defining the PRDS property (Benjamini and Yekutieli, 2001).

**Definition S1.** *A set $\mathcal{D}$ is called non-decreasing if $x \in \mathcal{D}$ and $x \leq y$ implies $y \in \mathcal{D}$.*

**Definition S2.** *For any non-decreasing set $\mathcal{D}$, and for each $i \in I_0$ such that $\mathbb{P}(X \in \mathcal{D}|X_i = x)$ is non-decreasing in $x$, then $X$ is PRDS on $I_0$.*

Theorem 3 states that the Storey conformal data contamination p-values are PRDS. This is a result motivated by Theorem 2.4 of Bates et al. (2023) stating that conformal p-values are PRDS. The implication of Theorem 3 is that we can use the BH procedure with the conformal data contamination p-values and maintain FDR control.

*Proof of Theorem 3.* Let $F$ be a shorthand for the function in Eq. (3), i.e., $\hat{u}_k^{\text{storey}} = F(\hat{p}_1^k, \dots, \hat{p}_{m_k}^k)$. It follows that $F$ is entry-wise monotone increasing (not strictly), i.e., $F(\hat{p}_1^k, \dots, (\hat{p}_j^k)', \dots, \hat{p}_{m_k}^k) \geq F(\hat{p}_1^k, \dots, \hat{p}_j^k, \dots, \hat{p}_{m_k}^k)$ where $(\hat{p}_j^k)' \geq \hat{p}_j^k$ for any $j \in \{1, \dots, m_k\}$. Let $Y = (\hat{u}_1^{\text{storey}}, \dots, \hat{u}_K^{\text{storey}})$ be the Storey conformal data contamination p-values on the test sets. Denote by $X_k = (\hat{p}_1^k, \dots, \hat{p}_{m_k}^k)$ the conformal p-values on the $k$-th test set, and by $X = (X_1, \dots, X_k)$ the total conformal p-values on the test sets.

Let $y \geq y'$ and let $A$ be an increasing set. Conditioning on the $k$-th conformal data contamination p-value and using the law of total probability

$$\mathbb{P}(Y \in A|Y_k = y) = \sum_{x \in \mathcal{X}} \frac{\mathbb{P}(Y \in A, Y_k = y, X_k = x)}{\mathbb{P}(Y_k = y)},$$

where $\mathcal{X} = \{1/(n+1), \dots, 1\}^m$. Now, by conditional independence of $Y_{-k} = (Y_1, \dots, Y_{k-1}, Y_{k+1}, \dots, Y_K)$ and $Y_k$

$$\mathbb{P}(Y \in A|Y_k = y) = \sum_{x \in \mathcal{X}} \mathbb{P}(Y \in A|X_k = x)\mathbb{P}(X_k = x|Y_k = y).$$

We know that $\mathbb{P}(Y_k|X_k = x) = \mathbb{1}[Y_k = F(x)]$, and so $\mathbb{P}(X_k|Y_k = y)$ is only non-zero when $y = F(x)$. Define $S(y) = \{x \in \mathcal{X} : y = F(x)\}$.

$$\mathbb{P}(Y \in A|Y_k = y) = \frac{\sum_{x \in S(y)} \mathbb{P}(X \in B|X_k = x)}{|S(y)|},$$

where $B = \{\bar{x} \in \mathcal{X}^K : [F(\bar{x}_1), \dots, F(\bar{x}_K)] \in A\}$. Since $A$ is an increasing set, and $F$ is an increasing function, $B$ is also an increasing set. By Theorem 2.4 of Bates et al. (2023), conformal p-values are PRDS, and as a corollary to this, we have that $\mathbb{P}(X \in B|X_k = x) \geq \mathbb{P}(X \in B|X_k = x')$ where $x \succeq x'$ ($\succeq$ denotes entry-wise inequality).

We define some notations: let $S^{\text{vec}}(y)$ denote a vectorization of the set $S(y)$ such that $S^{\text{vec}}(y) = (\tilde{x}_1^{S(y)}, \dots, \tilde{x}_{|S(y)|}^{S(y)})$ with $\tilde{x}_i^{S(y)} \in S(y)$, $\tilde{x}_i^{S(y)} \neq \tilde{x}_j^{S(y)}$, $i \neq j$, $i, j \in [|S(y))|]$; let also $\bar{S}^{\text{vec}}(y)$ denote the vectorization of $\bar{S}(y) \subset S(y)$; finally let $S^{\text{c}}(y)$ denote the complement of $S(y)$.

Assume initially that $|S(y')| = |S(y)|$. Then, $S(y)$ dominates $S(y')$ since $F$ is an entry-wise increasing function, i.e., there exists a permutation $\sigma$ on $[|S(y))|]$ such that $(\tilde{x}_{\sigma(1)}^{S(y)}, \dots, \tilde{x}_{\sigma(|S(y)|)}^{S(y)}) \succeq (\tilde{x}_1^{S(y')}, \dots, \tilde{x}_{|S(y)|}^{S(y')})$, and hence,

$$\mathbb{P}(Y \in A|Y_k = y) \geq \frac{\sum_{x' \in S(y')} \mathbb{P}(X \in B|X_k = x')}{|S(y')|} = \mathbb{P}(Y \in A|Y_k = y').$$

Now for the case $|S(y)| > |S(y')|$. Let $\bar{S}(y)$ be the subset of $S(y)$ with cardinality $|S(y')|$ which dominates $S(y')$ while minimizing $\sum_{x \in \bar{S}(y)} \mathbb{P}(X \in B|X_k = x)$, i.e., there exists a permutation $\sigma$ on $[|S(y')|]$ such that $(\tilde{x}^{\bar{S}(y)}_{\sigma(1)}, \ldots, \tilde{x}^{\bar{S}(y)}_{\sigma(|S(y')|)}) \succeq (\tilde{x}^{S(y')}_1, \ldots, \tilde{x}^{S(y')}_{|S(y')|})$, and hence,

$$
\begin{aligned}
\mathbb{P}(Y \in A|Y_k = y) &= \frac{\sum_{x \in \bar{S}(y)} \mathbb{P}(X \in B|X_k = x) + \sum_{x \in \bar{S}^c(y)} \mathbb{P}(X \in B|X_k = x)}{|S(y')| + |\bar{S}^c(y)|} \\
&\geq \frac{\sum_{x \in \bar{S}(y)} \mathbb{P}(X \in B|X_k = x)}{|S(y')|} \\
&\geq \frac{\sum_{x' \in S(y')} \mathbb{P}(X \in B|X_k = x)}{|S(y')|} \\
&= \mathbb{P}(Y \in A|Y_k = y').
\end{aligned}
$$

Finally for the case $|S(y')| > |S(y)|$. Let $\bar{S}(y')$ be the subset of $S(y')$ with cardinality $S(y)$ which is dominated by $S(y)$ while maximizing $\sum_{x' \in \bar{S}(y')} \mathbb{P}(X \in B|X_k = x')$, i.e., there exists a permutation $\sigma$ on $[|S(y)|]$ such that $(\tilde{x}^{\bar{S}(y')}_{\sigma(1)}, \ldots, \tilde{x}^{\bar{S}(y')}_{\sigma(|S(y)|)}) \preceq (\tilde{x}^{S(y)}_1, \ldots, \tilde{x}^{S(y)}_{|S(y)|})$. Now observe that

$$
\begin{aligned}
\mathbb{P}(Y \in A|Y_k = y) &= \frac{\sum_{x \in S(y)} \mathbb{P}(X \in B|X_k = x)}{|S(y)|} \\
&\geq \frac{\sum_{x' \in \bar{S}(y')} \mathbb{P}(X \in B|X_k = x')}{|S(y)|} \\
&\geq \frac{\sum_{x' \in \bar{S}(y')} \mathbb{P}(X \in B|X_k = x') + \sum_{x' \in \bar{S}^c(y')} \mathbb{P}(X \in B|X_k = x')}{|S(y')|} \\
&= \mathbb{P}(Y \in A|Y_k = y').
\end{aligned}
$$

The existence of such $\bar{S}(y)$ and $\bar{S}(y')$ is confirmed for the Storey conformal data contamination p-value. Consider initially that the Storey test statistic, denoted by $T$ here, is $T = t + 1$ and $T' = t$, i.e., just a difference of one. Now, $S(y) = \{x \in \mathcal{X} : x_{(1)} \leq \cdots \leq x_{(K-t-1)} \leq \lambda < x_{(K-t)} \leq \cdots \leq x_{(K)}\}$, and $S(y') = \{x' \in \mathcal{X} : x'_{(1)} \leq \cdots \leq x'_{(K-t)} \leq \lambda < x'_{(K-t+1)} \leq \cdots \leq x'_{(K)}\}$, and so the only difference occurs at $x_{(K-t)}$ which for $y$ is greater than or equal to $\lambda$ while for $y'$ it is less than or equal to $\lambda$. Now, for the case of $|S(y)| > |S(y')|$, for any $x' \in S(y')$ we can find an $x \in S(y)$ such that $x \succeq x'$. Taking for each $x' \in S(y')$ a unique $x \in S(y)$ with smallest $\mathbb{P}(X \in B|X_k = x)$ (avoiding using the same $x$ more than once), we have constructed $\bar{S}(y)$. In the same way we can construct $\bar{S}(y')$, taking for each $x \in S(y)$ a unique $x' \in S(y')$ with the largest $\mathbb{P}(X \in B|X_k = x')$. Extending the argument to differences in the Storey test statistic of more than one is immediate. $\square$

We have only proven the PRDS property for the Storey conformal data contamination p-values in Theorem 3, however, we expect the result can also be shown for the other conformal data contamination p-values.

## S2 ADDITIONAL DISCUSSION OF THE PROPOSED DATA SHARING PROCEDURE

In this section, we provide additional discussions of the proposed data sharing procedure. We summarize the data sharing procedure described in Section 4 in Procedure 1. We present an overview of the scenario variables and hyperparameters with some important interpretations in S2.1.

### S2.1 INTERPRETATIONS OF SCENARIO VARIABLES AND HYPERPARAMETERS

We outline in Tables S1 and S2 all scenario variables and hyperparameters. Here we recall the notation for each variable, and discuss how it influences the collaborative data sharing method proposed in this work.

**Procedure 1**

Input: local data $\{Z_i\}_{i=1}^{\ell}$ and $\{Z_i\}_{i=\ell+1}^{n}$, conformal score method $\hat{s}$, model class $f$, incoming data per round per data agent $m$, collaboration budget $K_{\text{budget}}$ (or contamination threshold $\pi_{\text{th}}$ and significance level $\alpha$), conformal non-contamination statistic $T$.

1: Fit conformal score $\hat{s}(\cdot, (Z_1, \ldots, Z_\ell))$, and compute $\hat{s}_{\ell+1}, \ldots, \hat{s}_n$.
2: Receive data (round 1) from other data agents $\{Z_i^k\}_{i=1}^{m}, k \in [K]$.
3: For each $k \in [K]$, compute conformal scores on the test data, $\hat{s}_1^k, \ldots, \hat{s}_m^k$, and subsequently conformal p-values, $\hat{p}_1^k, \ldots, \hat{p}_m^k$, using Eq. (2).
4: Evaluate conformal non-contamination statistics $T_k = T(\hat{p}_1^k, \ldots, \hat{p}_m^k)$ and find the corresponding conformal data contamination p-values, $\hat{u}_k$ (see Section 3).
5: **if** Given a fixed collaboration budget **then**
6:     Select for collaboration in the following round data agents in $\hat{\mathcal{H}}_0 = \{\sigma(i) : i \in [K_{\text{budget}}]\}$, where $\sigma$ is a permutation on $[K]$ such that $T_{\sigma(1)} \geq T_{\sigma(2)} \geq \cdots \geq T_{\sigma(K)}$, and estimate the FDR for null hypotheses $H_0^k : \pi_k \leq \pi_{\text{th}}$ for a $\pi_{\text{th}}$ of interest with $\hat{u}_k, k \in [K]$, using Eq. (1).
7: **else**
8:     Collaborate in the following round with data agents in $\hat{\mathcal{H}}_0 = [K] \setminus \text{SBH}_{\alpha,\gamma}(\hat{u}_1, \ldots, \hat{u}_K)$.
9: **end if**
10: Receive data (round 2, $\ldots$) from other data agents $\{Z_{m+i}^k\}_{i=1}^{m}, k \in \hat{\mathcal{H}}_0$.
11: Run data subset selection on all the received data.
12: Use all local data $\{Z_i\}_{i=1}^{n}$ together with the selected data to train the model, yielding $f^*$.

Output: optimized local model, $f^*$.

Table S1: Scenario variables.

| Parameter | Description & Influence |
| --- | --- |
| $n$ | The size of the null sample. As $n - \ell$ increases, the empirical distribution better approximates the true distribution, yielding more accurate conformal p-values. Moreover, this variable controls the smallest possible p-value, thereby controlling how much evidence one outlier can yield towards rejection in the conformal data contamination tests. |
| $m$ | The number of test data points. As this variable increases, more evidence against the null can be aggregated yielding a more powerful conformal data contamination test. |
| $\pi$ | The true contamination factor. As the difference $\pi - \pi_{\text{th}}$ increases, the power of the conformal data contamination tests increases. |
| $K$ | The number of other data agents. |
| $K_0$ | The number of other data agents satisfying the null hypothesis $H_0^k : \pi_k \leq \pi_{\text{th}}$. As this grows, the power of the multiple testing procedure decreases. |

## S3 ADDITIONAL NUMERICAL EXPERIMENTS: GAUSSIAN DATA

In this section, we present a simulation study with null distribution $P_0 \equiv \mathcal{N}(\mathbf{0}_2, \boldsymbol{I}_{2\times 2})$ and alternative $P_1 \equiv \mathcal{N}(\mu_1 \mathbf{1}_2, \boldsymbol{I}_{2\times 2})$. As conformal score we use $\hat{s}(X) = -\|X\|$ as it is a natural choice in light of the distribution $P_0$. We simulate $n = 200$ data points from $P_0$ to be the calibration data set and let the test data consist of $m = 50$ data points. The test data on average consists of $m(1 - \pi)$ data points sampled from $P_0$ while the remaining data points are sampled from $P_1$.

### S3.1 HYPERPARAMETERS OF THE CONFORMAL DATA CONTAMINATION TESTS

We compare, in Figure S1, the power estimated using 10000 simulations when $\alpha = 0.05$, $\mu_1 = 4$, $\pi = 0.7$, $\pi_{\text{th}} = 0.5$, $n = 200$, and $m = 50$, depending on the hyperparameters of the conformal data contamination tests, for the two cases $\mu_1 = 2$ and $\mu_1 = 4$. We can observe that for appropriate hyperparameter choices, the Storey and Quantile conformal data contamination tests can achieve the highest power. Moreover, when the outliers are easily separated from the inliers ($\mu_1 = 4$), $\lambda$ should be chosen as a very small value and the choice of $i_0$ should also be sufficiently small, meanwhile for $\mu_1 = 2$, $\lambda$ and $i_0$ should be chosen moderately. This highlights a weakness of the Storey and Quantile tests as they are sensitive to the hyperparameter choice. Note also that the Fisher test is

Table S2: Hyperparameters.

| | |
|---|---|
| $\ell$ | The number of null data points used for learning the conformal score. As this variable increases the conformal score will tend to better separate outliers from inliers thereby improving outlier detection and increasing power of the conformal data contamination tests. |
| $\hat{s}$ | The conformal score function. Choosing a good conformal score is key to achieving good separation between outliers and inliers, which also directly impacts the power and tightness of the conformal data contamination tests. |
| $K_{\text{budget}}$ | The collaboration budget, i.e., the number of other data agents to communicate with in the second round (in case a fixed collaboration budget is used). |
| $\pi_{\text{th}}$ | The chosen threshold on the contamination factor defining the null hypotheses. This hyperparameter controls the amount of contamination that is tolerated. |
| $\alpha$ | The significance level for the conformal data contamination tests. This hyperparameter controls the amount of evidence needed before concluding that the data from another agent is more contaminated than the threshold $\pi_{\text{th}}$ allows. |
| $\gamma$ | The hyperparameter in Storey's BH procedure. Controls the bias-variance trade-off for estimating $K_0$. |
| $\lambda$ | The hyperparameter in the Storey test statistic. |
| $i_0$ | The hyperparameter in the Quantile test statistic. |

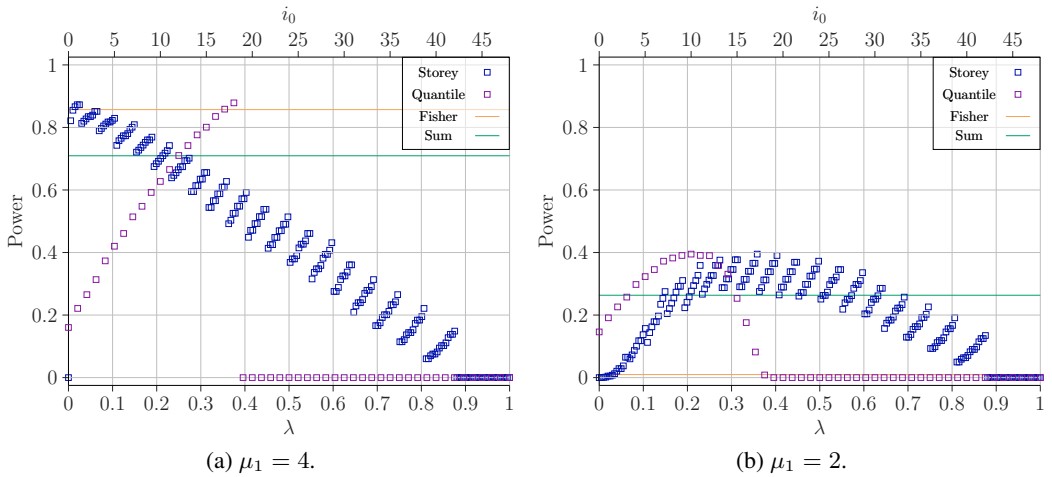

(a) $\mu_1 = 4$.           (b) $\mu_1 = 2$.

Figure S1: Comparison of power of the different tests for varying hyperparameters.

relatively powerful when the outliers are easily separated, however it is very weak when this is not the case. This is a sensible observation when considering the function used in the Fisher test statistic as it emphasizes the information in very small p-values, but when the outliers are not easily separated, the p-values for the outliers will not be very small and so is less emphasized.

### S3.2 SPECIAL CASE OF CLASSICAL TWO-SAMPLE TESTING

In the particular case of $\pi_{\text{th}} = 0$, the problem reverts to the classical two-sample testing problem. This can be solved using classical techniques such as Kolmogorov-Smirnov and Cramér-von Mises, but also permutation tests as in Konstantinou et al. (2024), as well as the combination tests of Bates et al. (2023). Notably, the Fisher conformal data contamination test proposed in this work coincides with the Fisher combination test of Bates et al. (2023) for $\pi_{\text{th}} = 0$.

The setup is the following: we observe two samples of data $\mathcal{D}_0 = \{X_i\}_{i=1}^n$ and $\mathcal{D}_1 = \{X_i\}_{i=n+1}^m$. Now, many different two-sample tests can be constructed by defining a summary statistic $\hat{s}(X)$, and then comparing the empirical distributions of the summary statistic on the two samples: $\hat{F}_0 = \hat{F}(\hat{s}(\mathcal{D}_0))$ and $\hat{F}_1 = \hat{F}(\hat{s}(\mathcal{D}_1))$, where $\hat{s}(\mathcal{D}_1) = \{\hat{s}(X_{n+i})\}_{i=1}^m$ and $\hat{F}$ computes the empirical CDF. Now, a test statistic can be constructed from these two empirical distributions $T(\hat{F}_0, \hat{F}_1)$.

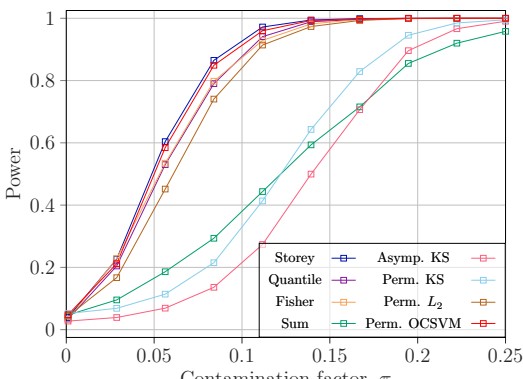

Figure S2: Comparison of the power of different two-sample tests for varying contamination factors.

**Kolmogorov-Smirnov test:**    A classic test statistic is the Kolmogorov-Smirnov statistic

$$T_{\mathrm{KS}}(\hat{F}_0, \hat{F}_1) = \sup_x |\hat{F}_0(x) - \hat{F}_1(x)|,$$

for which asymptotically as $n, m \to \infty$

$$\mathbb{P}\left(T_{\mathrm{KS}}(\hat{F}_0, \hat{F}_1) > \sqrt{-\ln(\alpha/2)\frac{n+m}{2nm}}\right) \le \alpha,$$

with significance level $\alpha \in (0, 1)$.

**Permutation tests:**    Permutation tests are based on constructing random permutations by randomly assigning data points from $\mathcal{D}_0$ and $\mathcal{D}_1$ to new splits $\mathcal{D}_{i,0}$ and $\mathcal{D}_{i,1}$ for $i = 1, \ldots I$ where $I$ is the number of permutations. The idea is that under the null, the test statistic $T(\hat{F}_0, \hat{F}_1)$ is exchangeable with $T_i = T(\hat{F}(\hat{s}(\mathcal{D}_{i,0})), \hat{F}(\hat{s}(\mathcal{D}_{i,1})))$ for $i = 1, \ldots I$ (Konstantinou et al., 2024). This allows for computing a p-value based on the empirical distribution of the test statistic among the permutations, allowing for flexibility in the choice of test statistic $T$, however, at a computational cost. We consider as examples test statistic using the Kolmogorov-Smirnov difference, the $L_2$-norm, and a test statistic using one class support vector machines (OCSVM). For the OCSVM, a number of training permutations are made yielding $T_1, \ldots, T_B$, followed by a number of calibration permutations $T_{B+1}, \ldots, T_I$. Then, the OCSVM is fitted to $T_1, \ldots, T_B$, and the output scores of the OCSVM on $T_{B+1}, \ldots, T_I$ are used a calibration data with the OCSVM score of $T(\hat{F}_0, \hat{F}_1)$ being the observed test statistic.

**Numerical power comparison:**    A power comparison across 2000 simulations of different two-sample tests for varying contamination factors when $\alpha = 0.05$, $n = 200$, $m = 100$, $\mu_1 = 4$, $\lambda = \lfloor n/32 \rfloor /(n + 1)$, and $i_0 = 5$, is shown in Figure S2. We have used 200 permutations for fitting OCSVM (in Perm. OCSVM), and 500 permutations for calibration (in the permutation tests). Figure S2 indicates that the weakest tests are the Kolmogorov-Smirnov tests and the Sum conformal data contamination test, and the other five tests achieve relatively similar power. We note that it intuitively makes sense that the Fisher combination test is powerful here as this test statistic emphasizes the contribution of very small p-values more so than the other test statistics, which is a desirable property when looking for a minimum of just one outlier. On the other hand, the Sum conformal data contamination test is relatively weak in this setting. These properties of these two tests were also discussed in the context of Figure S1. Our main take-away is that the proposed conformal data contamination tests are competitive with state-of-the-art tests.

### S3.3    ABLATION STUDY: TEST AND CALIBRATION DATA SIZE

In this section we study the performance of the conformal data contamination tests for varying parameter settings. Specifically, the focus will be on varying contamination factors, $\pi$, calibration data sizes, $n - \ell$, and test data sizes, $m$. We set $\lambda = \lfloor n/12 \rfloor /(n + 1)$, $i_0 = \lfloor m/1.5 \rfloor$, $\alpha = 0.05$, $\pi_{\mathrm{th}} = 0.1$, and $\mu_1 = 4$.

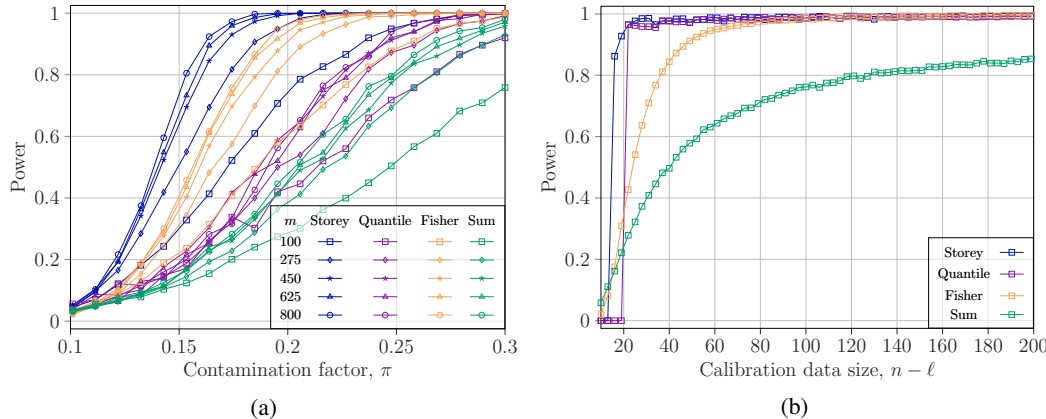

Figure S3: Ablation study for varying contamination factors, $\pi$, calibration data sizes, $n - \ell$, and test data sizes, $m$.

Consider an ablation study for the contamination factor, $\pi$, and the test data size, $m$, with $n = 100$. We show in Figure S3a the power estimated using 1000 simulations for the conformal data contamination tests. We observe that the power relatively quickly goes to 1 as the contamination factor increases, and there is a general tendency that the power increases with $m$, which is also to be expected. In this simulation study, the Storey conformal data contamination test achieves the highest power.

The influence of the calibration data size, $n - \ell$, with $m = 100$ and $\pi = 0.3$, is visualized in Figure S3b. Here we show the power estimated using 10000 simulations for the conformal data contamination tests. The Storey and Quantile conformal data contamination tests have the highest power, achieving a power near 1 for $n - \ell \geq 20$. This highlights that powerful conformal data contamination tests are possible even for extremely modest calibration data sizes.

With the insights gained through these numerical experiments we can highlight an advantage of the conformal data contamination tests. Conformal outlier detection relies on having a calibration data size, $n - \ell$, relatively large compared to the test data size, $m$, see Mary and Roquain (2022). Meanwhile, conformal data contamination tests only improve with increasing $m$, and so are more convenient in settings with small $n - \ell$ and large $m$.

### S3.4 MULTIPLE TESTING WITH CONFORMAL DATA CONTAMINATION P-VALUES

Consider the same scenario as before except now $K = 20$ agents are sending test data for which $K_0 = 10$ of them have $\pi_0 = \pi_{\text{th}}$ and the remaining $K - K_0 = 10$ have $\pi_1 > \pi_{\text{th}}$. We use Storey's BH procedure on the sequence of conformal data contamination p-values with Storey's hyperparameter $\gamma = 0.5$.

In Figure S4 we show FDR and TDR curves estimated by 2000 simulations for the four different conformal data contamination tests presented in Section 3, when $\alpha = 0.05$, $\mu_1 = 4$, $n = 200$, $m = 100$, $\pi_{\text{th}} = 0.2$, $\pi_1 = 0.3$, $\lambda = \lfloor n/32 \rfloor / (n + 1)$, and $i_0 = \lfloor m/1.5 \rfloor$. We notice from Figure S4a that all the methods are conservative. In terms of power Figure S4b shows that the Storey conformal data contamination test achieves the highest power, with comparable power between Quantile and Fisher test, and the Sum test tends to have the lowest power. We emphasize that this does not mean that in general the Sum test is the worst alternative among those considered here. Which of the proposed tests performs the best depends on many factors, herein $\pi_{\text{th}}$ and the conformal score.

## S4 ADDITIONAL NUMERICAL EXPERIMENTS: REAL DATASETS

In this section, we provide extensive numerical results on a variety of data contamination scenarios with real image classification datasets. We consider several conformal scores and evaluate both the outlier detection capability, the performance of the proposed conformal data contamination tests, and

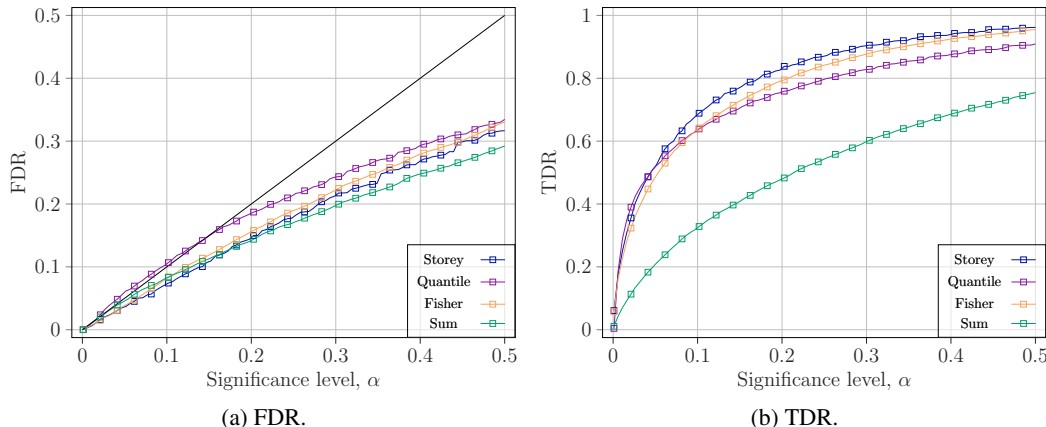

Figure S4: Comparison of FDR and TDR estimated of the different tests.

the performance on the relevant classification task for the proposed data acquisition procedure as well as relevant baselines.

In the following we list the data contamination scenarios.

(S1) Mixture of retinal fundus images (color) from RetinaMNIST (Liu et al., 2022) and EyePACS (Dugas et al., 2015; Gulshan et al., 2016), as also considered in Section 5. The classification task is to distinguish between five degrees of severity of diabetic retinopathy ranging from *no diabetic retinopathy* to *proliferate retinopathy*.

(S2) Mixture of optical coherence tomography (OCT) images (grayscale) taken from University of California San Diego found in OCTMNIST (Kermany et al., 2018) and OCT images from Noor Eye Hospital (NEH) (Sotoudeh-Paima et al., 2022). The classification task is to distinguish *choroidal neovascularization* from *normal retina*.

(S3) Mixture of microscopic blood smear images (color) from BloodMNIST (Acevedo et al., 2020) and (Bodzas et al., 2023). The classification task to to distinguish between the labels *basophile*, *eosinophile*, *lymphocyte*, and *monocyte*.

(S4) Mixture of images (grayscale) of lower- and uppercase handwritten letters *A*, *B*, and *C* from FEMNIST (Caldas et al., 2019).

(S5) Mixture of noise-free and noisy images (grayscale) of handwritten digits 1, 4, and 7 from MNIST (Lecun et al., 1998). We sample the feature noise as a Gaussian vector with zero mean and covariance kernel $\varsigma^2 \exp(-\varrho\|p_i - p_j\|)$ for pixel positions $p_i = i$ and $p_j = j$ for the $i$-th row and $j$-column in the pixel image, and add it to the image. After adding noise, each image is normalized to the interval $[0, 1]$. We set $\varsigma = 0.3$ and $\varrho = 0.1$.

(S6) Mixture of images (grayscale) of handwritten digits 1, 4, and 7 from MNIST with and without noisy labels.

(S7) Mixture of images of dogs, deer, and cats from CIFAR-10 and CIFAR-10C contaminated with brightness (Hendrycks and Dietterich, 2019).

(S8) Mixture of images of dogs, deer, and cats from CIFAR-10 and CIFAR-10C contaminated with snow (Hendrycks and Dietterich, 2019).

The images in (S1), (S2), and (S3) are cropped and down sampled in the same way to $28 \times 28$ ($28 \times 28 \times 3$ for color images) pixels. All images are pixel-wise normalized to the interval $[0, 1]$.

In all cases we sample randomly $n$ in-distribution datapoints for the 0-th data agent. Meanwhile, the $k$-th data agent observes $2m$ datapoints and the number of OOD datapoints is sampled from $\mathrm{B}_{\pi_k}^{2m}$. We sample the contamination factors iid according to a standard uniform distribution, such that approximately half of the test data are outliers. For all the numerical experiments we set $n = 100$, $\ell = 60$, $m = 40$, $K = 10$, $\lambda = \lfloor n/8 \rfloor/(n + 1)$, $i_0 = \lfloor m/3 \rfloor$, and $\gamma = 0.5$. Here we chose $n$ relatively small as this is the regime where data sharing is most interesting, and we decided on a 60-40 split between data for fitting conformal scores and calibration data, as we found that having

enough data for fitting the conformal score is often critical. The hyperparameters $\lambda$ and $i_0$ were chosen in light of the insights gained in the ablation study in Section S3. The hyperparameter for Storey's BH procedure, $\gamma$, was chosen somewhat arbitrarily at the intermediate value $0.5$.

We consider different conformal scores:

(C1)  OCSVM (Schölkopf et al., 2001).

(C2)  Isolation forest (IF) (Liu et al., 2008).

(C3)  Autoencoder (AE) (Hawkins et al., 2002).

(C4)  PU classification using LR (Marandon et al., 2024).

(C5)  PU classification using SVC (Marandon et al., 2024).

(C6)  Class-wise score based on OCSVM.

(C7)  Class-wise score based on IF .

(C8)  Class-wise score based on AE.

(C9)  Class-wise score based on PU classification using LR.

(C10)  Class-wise score based on PU classification using SVC.

(C11)  PU classification using LR on principal components explaining at least $80\,\%$ of the variance.

(C12)  Class-wise score based on PU classification using LR on principal components explaining at least $80\,\%$ of the variance.

Class-wise scores are defined as

$$\hat{s}((\boldsymbol{X}, Y), (Z_1, \ldots, Z_\ell), (Z_{\ell+1}, \ldots, Z_{n+m})) = \sum_{i \in \mathcal{Y}} \mathbb{1}[Y = i] \, \hat{s}_i(\boldsymbol{X}, \boldsymbol{X}_{1:\ell}^{Y=i}, \boldsymbol{X}_{\ell+1:n+m}^{Y=i}),$$

where $\mathcal{Y}$ denotes the set of labels, $\boldsymbol{X}_{1:\ell}^{Y=i} = (\boldsymbol{X}_j : Y_j = i, j \in \{1, \ldots, \ell\})$ denotes the subset of $\boldsymbol{X}_1, \ldots, \boldsymbol{X}_\ell$ for which $Y_j = i$ for $j \in \{1, \ldots, \ell\}$.

All scores are implemented using the standard implementations in `scikit-learn`, except the AEs which are implemented in `tensorflow`.

### S4.1   Marginal power and type I error probability study

To compare the different conformal scores on the various data contamination scenarios we will consider the marginal power of the conformal outlier detection and conformal data contamination tests, as well as the type I error probability for the conformal data contamination tests, through the empirical CDFs of the conformal p-values and the conformal data contamination p-values, respectively. Particularly, we will use the area under the curve (AUC) to summarize the CDFs to scalars.

For conformal outlier detection, $\mathrm{AUC}^{\mathrm{cod}} = 1/(n+2) \sum_{i=1}^{n+1} \hat{F}_{\mathrm{outlier}}(i/(n+1))$ where $\hat{F}_{\mathrm{outlier}}$ is the empirical CDF of the conformal p-value among outliers. $\mathrm{AUC}^{\mathrm{cod}}$ ranges from 0 to 1 with 0.5 corresponding to random guessing and 1 corresponding to perfect detection.

For conformal data contamination tests, $\mathrm{AUC}^{cdct}_{(\pi, \pi_{\mathrm{th}})} = \int_0^1 \hat{F}_{(\pi, \pi_{\mathrm{th}})}(x)\mathrm{d}x$ where $\hat{F}_{(\pi, \pi_{\mathrm{th}})}$ is the empirical CDF of the conformal data contamination test p-value for data contamination threshold $\pi_{\mathrm{th}}$ and when the data contamination level is $\pi$. For comparing the power we are interested in $\pi > \pi_{\mathrm{th}}$ and a larger $\mathrm{AUC}^{cdct}_{(\pi, \pi_{\mathrm{th}})}$ the better, while for numerically evaluating the conservativeness we are interested in $\pi = \pi_{\mathrm{th}}$ which should be less than but also close to $0.5$.

We report in Table S3 $\mathrm{AUC}^{\mathrm{cod}}$ for the considered data contamination scenarios and for each of the different conformal scores, and show the full CDFs in Figure S5. Immediately, we can notice that the adaptive scores using the PU classification techniques of Marandon et al. (2024) consistently perform well. For all the medical data scenarios (S1), (S2), and (S3), AdaDetect with SVC, i.e. (C5), as well as the AE, i.e. (C3), perform very well. For scenarios (S4), (S5), and (S6), class-wise scores generally perform well with AdaDetect using LR, i.e. (C9), consistently yielding accurate outlier detection; however, for (S5) the best score, by a significant margin, is AdaDetect using LR on the

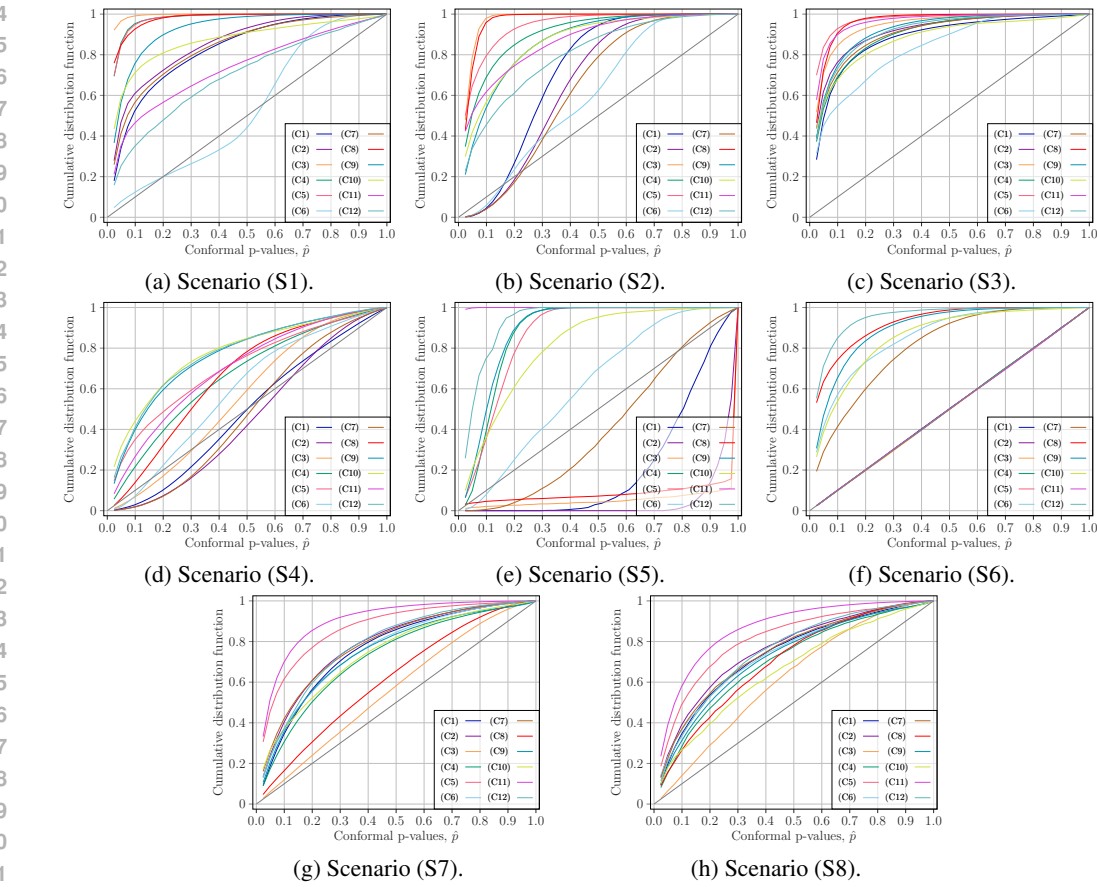

Figure S5: *Power study:* Empirical CDFs of $\text{AUC}^{\text{cod}}$ for the different conformal scores and data contamination scenarios.

Table S3: *Power study:* AUC for conformal outlier detection computed from 2000 simulations. The best conformal score for each data contamination scenario is highlighted with boldface.

|        | (S1)   | (S2)   | (S3)   | (S4)   | (S5)   | (S6)   | (S7)   | (S8)   |
|--------|--------|--------|--------|--------|--------|--------|--------|--------|
| (C1)   | 0.8164 | 0.7110 | 0.8694 | 0.4819 | 0.2246 | 0.5011 | 0.7531 | 0.7279 |
| (C2)   | 0.8439 | 0.6582 | 0.9003 | 0.4539 | 0.0586 | 0.4996 | 0.7738 | 0.7501 |
| (C3)   | **0.9726** | **0.9562** | 0.9287 | 0.5554 | 0.0712 | 0.4984 | 0.5594 | 0.6116 |
| (C4)   | 0.9584 | 0.8921 | 0.8863 | 0.6580 | 0.8676 | 0.4999 | 0.7164 | 0.6948 |
| (C5)   | 0.9580 | 0.9259 | **0.9537** | 0.6979 | 0.8525 | 0.4992 | 0.8543 | 0.7999 |
| (C6)   | 0.5343 | 0.6047 | 0.8240 | 0.5852 | 0.6158 | 0.8451 | 0.7504 | 0.7292 |
| (C7)   | 0.8261 | 0.6333 | 0.8894 | 0.4773 | 0.4077 | 0.7979 | 0.7734 | 0.7361 |
| (C8)   | 0.9581 | 0.9515 | 0.9450 | 0.6569 | 0.0997 | **0.9088** | 0.6092 | 0.6913 |
| (C9)   | 0.9105 | 0.8536 | 0.9058 | 0.7572 | 0.8792 | 0.8850 | 0.7409 | 0.7124 |
| (C10)  | 0.8571 | 0.8549 | 0.8645 | **0.7678** | 0.8019 | 0.8438 | 0.7345 | 0.6530 |
| (C11)  | 0.7283 | 0.8568 | 0.9435 | 0.6903 | **0.9759** | 0.4995 | **0.8896** | **0.8502** |
| (C12)  | 0.6904 | 0.7917 | 0.8982 | 0.7633 | 0.9223 | 0.9357 | 0.7725 | 0.7401 |

principal components. Meanwhile, the most effective score for the CIFAR-10 scenarios (S7) and (S8) is AdaDetect using LR on the principal components, i.e. (C11). The most challenging data contamination scenario here is (S4), i.e., the FEMNIST data scenario, meanwhile the apparently easiest is (S1), i.e., the retinal fundus image data scenario.

While these results for varying data contamination scenarios with different conformal scores are not exhaustive, they give an indication of the achieveable outlier detection performance, and provide some guidelines for choosing conformal scores. In the following, we will limit our attention to AdaDetect with SVC, i.e (C5), and class-wise AdaDetect with LR, i.e. (C9), as these scores perform consistently well, and we will intermediately restrict our attention to scenarios (S1)-(S6).

Table S4: *Power study:* AUC for conformal data contamination tests computed from 2000 simulations when $\pi = 0.3$. The best test statistic for each data contamination scenario and conformal score is highlighted with boldface.

| | | (S1) | | | (S2) | | | (S3) | | |
| --- | --- | --- | --- | --- | --- | --- | --- | --- | --- | --- |
| | | $\pi_{\mathrm{th}}=0$ | $\pi_{\mathrm{th}}=0.1$ | $\pi_{\mathrm{th}}=0.2$ | $\pi_{\mathrm{th}}=0$ | $\pi_{\mathrm{th}}=0.1$ | $\pi_{\mathrm{th}}=0.2$ | $\pi_{\mathrm{th}}=0$ | $\pi_{\mathrm{th}}=0.1$ | $\pi_{\mathrm{th}}=0.2$ |
| (C5) | Storey | **0.9796** | **0.9076** | **0.7251** | **0.9492** | **0.8250** | 0.6008 | **0.9790** | **0.9037** | **0.7135** |
| | Quantile | 0.8094 | 0.7371 | 0.6371 | 0.8172 | 0.7453 | **0.6442** | 0.8076 | 0.7356 | 0.6351 |
| | Fisher | 0.9664 | 0.8485 | 0.6087 | 0.9379 | 0.7719 | 0.4956 | 0.9674 | 0.8533 | 0.6105 |
| | Sum | 0.9220 | 0.8070 | 0.6206 | 0.9079 | 0.7843 | 0.5890 | 0.9198 | 0.8062 | 0.6203 |
| (C9) | Storey | **0.9514** | **0.8233** | 0.5867 | **0.8607** | 0.6470 | 0.3857 | **0.9462** | **0.8108** | 0.5720 |
| | Quantile | 0.8055 | 0.7309 | **0.6279** | 0.7969 | **0.7185** | **0.6110** | 0.7998 | 0.7244 | **0.6207** |
| | Fisher | 0.9289 | 0.7351 | 0.4456 | 0.8474 | 0.5762 | 0.2815 | 0.9245 | 0.7321 | 0.4446 |
| | Sum | 0.8987 | 0.7634 | 0.5632 | 0.8575 | 0.6952 | 0.4775 | 0.8959 | 0.7589 | 0.5558 |
| | | (S4) | | | (S5) | | | (S6) | | |
| | | $\pi_{\mathrm{th}}=0$ | $\pi_{\mathrm{th}}=0.1$ | $\pi_{\mathrm{th}}=0.2$ | $\pi_{\mathrm{th}}=0$ | $\pi_{\mathrm{th}}=0.1$ | $\pi_{\mathrm{th}}=0.2$ | $\pi_{\mathrm{th}}=0$ | $\pi_{\mathrm{th}}=0.1$ | $\pi_{\mathrm{th}}=0.2$ |
| (C5) | Storey | **0.7401** | 0.4551 | 0.2117 | 0.5688 | 0.4136 | 0.3236 | **0.5073** | 0.2124 | 0.0639 |
| | Quantile | 0.6694 | **0.5792** | **0.4676** | 0.8002 | **0.7259** | **0.6238** | 0.5027 | **0.4058** | **0.3005** |
| | Fisher | 0.6826 | 0.3698 | 0.1397 | 0.7538 | 0.4530 | 0.2083 | 0.4123 | 0.1407 | 0.0319 |
| | Sum | 0.6917 | 0.4917 | 0.2894 | **0.8439** | 0.6765 | 0.4617 | 0.4494 | 0.2496 | 0.1097 |
| (C9) | Storey | **0.7866** | 0.5165 | 0.2594 | 0.7636 | 0.5989 | 0.4263 | **0.9140** | **0.7367** | 0.4780 |
| | Quantile | 0.7206 | **0.6316** | **0.5175** | 0.8170 | **0.7451** | **0.6445** | 0.7910 | 0.7140 | **0.6091** |
| | Fisher | 0.7514 | 0.4292 | 0.1704 | 0.8247 | 0.5416 | 0.2606 | 0.8954 | 0.6594 | 0.3585 |
| | Sum | 0.7668 | 0.5674 | 0.3485 | **0.8786** | 0.7295 | 0.5191 | 0.8759 | 0.7245 | 0.5123 |

Table S5: *Type I error probability study:* AUC for conformal data contamination tests computed from 2000 simulations when $\pi = \pi_{\mathrm{th}}$.

| | | (S1) | | | (S2) | | | (S3) | | |
| --- | --- | --- | --- | --- | --- | --- | --- | --- | --- | --- |
| | | $\pi_{\mathrm{th}}=0$ $\pi=0$ | $\pi_{\mathrm{th}}=0.1$ $\pi=0.1$ | $\pi_{\mathrm{th}}=0.2$ $\pi=0.2$ | $\pi_{\mathrm{th}}=0$ $\pi=0$ | $\pi_{\mathrm{th}}=0.1$ $\pi=0.1$ | $\pi_{\mathrm{th}}=0.2$ $\pi=0.2$ | $\pi_{\mathrm{th}}=0$ $\pi=0$ | $\pi_{\mathrm{th}}=0.1$ $\pi=0.1$ | $\pi_{\mathrm{th}}=0.2$ $\pi=0.2$ |
| (C5) | Storey | 0.5032 | 0.4788 | 0.4638 | 0.5007 | 0.4229 | 0.3780 | 0.4911 | 0.4747 | 0.4545 |
| | Quantile | 0.5035 | 0.4995 | 0.5048 | 0.5019 | 0.4975 | 0.4944 | 0.4841 | 0.4888 | 0.4828 |
| | Fisher | 0.4217 | 0.3761 | 0.3522 | 0.4125 | 0.3334 | 0.2703 | 0.4002 | 0.3749 | 0.3399 |
| | Sum | 0.4587 | 0.4304 | 0.4151 | 0.4499 | 0.4161 | 0.3849 | 0.4352 | 0.4272 | 0.3959 |
| (C9) | Storey | 0.4906 | 0.4165 | 0.3537 | 0.4990 | 0.3358 | 0.2402 | 0.4934 | 0.4069 | 0.3517 |
| | Quantile | 0.4796 | 0.4932 | 0.4908 | 0.5035 | 0.5033 | 0.4832 | 0.5150 | 0.4987 | 0.4933 |
| | Fisher | 0.3843 | 0.3112 | 0.2299 | 0.4110 | 0.2537 | 0.1565 | 0.4106 | 0.3033 | 0.2383 |
| | Sum | 0.4184 | 0.4082 | 0.3591 | 0.4519 | 0.3829 | 0.3168 | 0.4545 | 0.4021 | 0.3669 |
| | | (S4) | | | (S5) | | | (S6) | | |
| | | $\pi_{\mathrm{th}}=0$ $\pi=0$ | $\pi_{\mathrm{th}}=0.1$ $\pi=0.1$ | $\pi_{\mathrm{th}}=0.2$ $\pi=0.2$ | $\pi_{\mathrm{th}}=0$ $\pi=0$ | $\pi_{\mathrm{th}}=0.1$ $\pi=0.1$ | $\pi_{\mathrm{th}}=0.2$ $\pi=0.2$ | $\pi_{\mathrm{th}}=0$ $\pi=0$ | $\pi_{\mathrm{th}}=0.1$ $\pi=0.1$ | $\pi_{\mathrm{th}}=0.2$ $\pi=0.2$ |
| (C5) | Storey | 0.5098 | 0.2662 | 0.1366 | 0.4875 | 0.2969 | 0.2303 | 0.5047 | 0.2119 | 0.0616 |
| | Quantile | 0.5031 | 0.4469 | 0.3983 | 0.4907 | 0.4962 | 0.4939 | 0.4955 | 0.4098 | 0.2936 |
| | Fisher | 0.4186 | 0.1926 | 0.0830 | 0.3938 | 0.2290 | 0.1284 | 0.4109 | 0.1464 | 0.0308 |
| | Sum | 0.4532 | 0.3124 | 0.2061 | 0.4298 | 0.3819 | 0.3168 | 0.4469 | 0.2559 | 0.1089 |
| (C9) | Storey | 0.5065 | 0.3045 | 0.1676 | 0.5011 | 0.3524 | 0.2746 | 0.5084 | 0.3888 | 0.3078 |
| | Quantile | 0.5068 | 0.4746 | 0.4374 | 0.5043 | 0.5022 | 0.4904 | 0.4996 | 0.5033 | 0.4930 |
| | Fisher | 0.4194 | 0.2168 | 0.1029 | 0.4078 | 0.2427 | 0.1412 | 0.4117 | 0.2900 | 0.2049 |
| | Sum | 0.4570 | 0.3441 | 0.2490 | 0.4492 | 0.3906 | 0.3333 | 0.4460 | 0.3997 | 0.3496 |

In Table S4, we display $\mathrm{AUC}^{\mathrm{cdct}}_{(\pi,\pi_{\mathrm{th}})}$, for $\pi > \pi_{\mathrm{th}}$ thereby indicating the power of the tests, for the four different proposed conformal data contamination tests for varying values of the data contamination threshold, $\pi_{\mathrm{th}}$, when the data contamination level is $\pi = 0.3$. The AUC decreases for increasing data contamination thresholds, which is to be expected as distinguishing between smaller differences in data contamination is naturally harder. We notice that in almost all cases, the Storey and Quantile tests, which are valid non-asymptotically, perform better than the Fisher and Sum tests, which are only valid asymptotically. Further, we observe that the Storey test usually has the highest AUC for small data contamination thresholds, whereas the Quantile test is comparatively better for larger data contamination thresholds. A similar observation holds when comparing the Fisher and Sum tests, with the Fisher test performing better at small data contamination thresholds, and vice versa, as to be expected.

Table S6: TDR computed from 2000 simulations at significance level $\alpha = 0.1$ using Storey's BH procedure with parameter $\gamma = 0.5$. Data contamination factors are simulated as $\pi_k \sim \mathrm{Uniform}([0,1])$. The best test statistic for each data contamination scenario and conformal score is highlighted with boldface.

| | | (S1) | | | (S2) | | | (S3) | | |
| | | $\pi_{\mathrm{th}}=0$ | $\pi_{\mathrm{th}}=0.1$ | $\pi_{\mathrm{th}}=0.2$ | $\pi_{\mathrm{th}}=0$ | $\pi_{\mathrm{th}}=0.1$ | $\pi_{\mathrm{th}}=0.2$ | $\pi_{\mathrm{th}}=0$ | $\pi_{\mathrm{th}}=0.1$ | $\pi_{\mathrm{th}}=0.2$ |
|---|---|---|---|---|---|---|---|---|---|---|
| (C5) | Storey | **0.9511** | **0.8984** | **0.8438** | 0.9135 | 0.8249 | 0.7172 | **0.9482** | **0.8914** | **0.8323** |
| | Quantile | 0.8236 | 0.8030 | 0.7863 | 0.8201 | 0.7976 | **0.7668** | 0.8152 | 0.7922 | 0.7738 |
| | Fisher | 0.9333 | 0.8629 | 0.7856 | 0.9029 | 0.7950 | 0.6687 | 0.9312 | 0.8601 | 0.7854 |
| | Sum | 0.9016 | 0.8464 | 0.7859 | 0.8873 | 0.8216 | 0.7310 | 0.8956 | 0.8410 | 0.7748 |
| (C9) | Storey | **0.9128** | **0.8147** | 0.7033 | 0.8014 | 0.5995 | 0.4145 | **0.9122** | **0.8056** | 0.6838 |
| | Quantile | 0.8104 | 0.7851 | **0.7537** | 0.7965 | **0.7548** | **0.6784** | 0.8105 | 0.7816 | **0.7407** |
| | Fisher | 0.8890 | 0.7576 | 0.6077 | 0.8076 | 0.5566 | 0.3298 | 0.8903 | 0.7539 | 0.6032 |
| | Sum | 0.8778 | 0.7994 | 0.7142 | **0.8400** | 0.7229 | 0.5821 | 0.8777 | 0.7970 | 0.7046 |

| | | (S4) | | | (S5) | | | (S6) | | |
| | | $\pi_{\mathrm{th}}=0$ | $\pi_{\mathrm{th}}=0.1$ | $\pi_{\mathrm{th}}=0.2$ | $\pi_{\mathrm{th}}=0$ | $\pi_{\mathrm{th}}=0.1$ | $\pi_{\mathrm{th}}=0.2$ | $\pi_{\mathrm{th}}=0$ | $\pi_{\mathrm{th}}=0.1$ | $\pi_{\mathrm{th}}=0.2$ |
|---|---|---|---|---|---|---|---|---|---|---|
| (C5) | Storey | **0.6295** | 0.2915 | 0.0955 | 0.4193 | 0.4068 | 0.3933 | **0.1095** | 0.0074 | 0.0000 |
| | Quantile | 0.5696 | **0.4308** | **0.2592** | 0.8130 | **0.7927** | **0.7752** | 0.1027 | **0.0347** | **0.0082** |
| | Fisher | 0.5850 | 0.2364 | 0.0649 | 0.6492 | 0.3669 | 0.2251 | 0.0589 | 0.0013 | 0.0000 |
| | Sum | 0.6288 | 0.3839 | 0.1711 | **0.8187** | 0.6909 | 0.5560 | 0.0816 | 0.0110 | 0.0001 |
| (C9) | Storey | 0.7028 | 0.3953 | 0.1939 | 0.6864 | 0.5398 | 0.4718 | **0.8784** | 0.7345 | 0.5799 |
| | Quantile | 0.6945 | **0.5850** | **0.4308** | 0.8213 | **0.8025** | **0.7791** | 0.8068 | **0.7770** | **0.7232** |
| | Fisher | 0.6867 | 0.3234 | 0.1142 | 0.7725 | 0.4868 | 0.2892 | 0.8671 | 0.6939 | 0.5033 |
| | Sum | **0.7468** | 0.5241 | 0.3013 | **0.8550** | 0.7569 | 0.6297 | 0.8675 | 0.7708 | 0.6634 |

In Table S5, we display $\mathrm{AUC}^{\mathrm{cdct}}_{(\pi, \pi_{\mathrm{th}})}$, for $\pi = \pi_{\mathrm{th}}$ thereby indicating the probability of false rejection, again for all the conformal data contamination tests for varying values of $\pi = \pi_{\mathrm{th}}$. As the developed theory promises, in all cases the AUC is less than (or approximately) $0.5$. These results are interesting to see the amount of excess conservativeness in the conformal data contamination tests, noting that an AUC of (approximately) $0.5$ means there is no excess conservativeness. Generally, the Quantile test shows the least amount of excess conservativeness with AUC approximately $0.5$ even when $\pi = \pi_{\mathrm{th}} = 0.2$ for most scenarios. Any excess conservativeness can in these results be attributed to the inequalities in Eqs. (8) and (13), in which an outlier with a conformal p-value above $1/(n+1)$ contributes to a lack of tightness in the bound.

## S4.2 MULTIPLE TESTING WITH CONFORMAL DATA CONTAMINATION P-VALUES

Now, we will investigate the TDR and FDR when doing multiple conformal data contamination tests. The TDR when using Storey's BH procedure at significance level $\alpha = 0.1$ and for Storey's hyperparameter $\gamma = 0.5$ is shown in Table S6. Mostly, the results follow that observed in Table S4 for the marginal power, i.e., the Storey test statistic tends to be the most powerful, but for some cases for larger data contamination thresholds, the Quantile test statistic is preferable. This tells us that the marginal power of the test is a good indication of the power of multiple testing.

To finally validate Theorem 3 on these real datasets, we show in Table S7 the FDR when using the BH procedure at level $\alpha = 0.1$ when at the boundary of the null hypothesis. Specifically, we let $K_0 = K/2$ of the data agents have $\pi_k = \pi_{\mathrm{th}}$ and the others have $\pi_k = 1 - \pi_{\mathrm{th}}$ (this maintains a balance of approximately half the data points being outliers). Hence, here we do not simulate $\pi_k \sim \mathrm{Uniform}([0,1])$ as we do not allow any $\pi_k < \pi_{\mathrm{th}}$ which would inherently cause excess conservativeness. By Theorem 3, the FDR should in this case be upper bounded by $\alpha K_0/K = 0.05$, which is also observed in Table S7 for all tested cases. However, we can note that some excess conservativeness arises when applying the BH procedure to the conformal data contamination p-values. Once again, the Quantile test is generally the least conservative and in some of the tested scenarios maintain an FDR close to the $0.05$ level even for $\pi_{\mathrm{th}} = 0.2$. On the other hand, the Fisher test tends to be the most conservative out of the four proposed conformal data contamination tests.

Finally, we consider, in Table S8, the FDR when using Storey's BH procedure. Here, you see that the tests are close to the nominal level $\alpha = 0.1$, even for $\pi_{\mathrm{th}} = 0.2$, especially the Quantile tests. Some conservativeness is induced from the multiplicity correction, as was reported above, however, using the adaptive procedure of Storey's BH tends to do the opposite. Recall that we have only shown FDR

Table S7: FDR computed from 2000 simulations at significance level $\alpha = 0.1$ using the BH procedure. For $K_0 = K/2$ of the data agents $\pi_k = \pi_{\text{th}}$ while for the remaining data agents $\pi_k = 1 - \pi_{\text{th}}$. Theoretically, the FDR is upper bounded by $\alpha K_0 / K = 0.05$.

| | | (S1) | | | (S2) | | | (S3) | | |
| | | $\pi_{\text{th}}=0$ | $\pi_{\text{th}}=0.1$ | $\pi_{\text{th}}=0.2$ | $\pi_{\text{th}}=0$ | $\pi_{\text{th}}=0.1$ | $\pi_{\text{th}}=0.2$ | $\pi_{\text{th}}=0$ | $\pi_{\text{th}}=0.1$ | $\pi_{\text{th}}=0.2$ |
|---|---|---|---|---|---|---|---|---|---|---|
| | Storey | 0.0394 | 0.0330 | 0.0347 | 0.0427 | 0.0270 | 0.0271 | 0.0390 | 0.0337 | 0.0359 |
| (C5) | Quantile | 0.0405 | 0.0434 | 0.0401 | 0.0433 | 0.0434 | 0.0372 | 0.0417 | 0.0435 | 0.0380 |
| | Fisher | 0.0149 | 0.0162 | 0.0145 | 0.0182 | 0.0133 | 0.0083 | 0.0154 | 0.0159 | 0.0140 |
| | Sum | 0.0296 | 0.0266 | 0.0228 | 0.0328 | 0.0268 | 0.0190 | 0.0286 | 0.0289 | 0.0223 |
| | Storey | 0.0363 | 0.0246 | 0.0251 | 0.0379 | 0.0197 | 0.0115 | 0.0414 | 0.0207 | 0.0234 |
| (C9) | Quantile | 0.0410 | 0.0391 | 0.0361 | 0.0447 | 0.0401 | 0.0324 | 0.0416 | 0.0403 | 0.0374 |
| | Fisher | 0.0164 | 0.0093 | 0.0067 | 0.0168 | 0.0066 | 0.0029 | 0.0193 | 0.0095 | 0.0052 |
| | Sum | 0.0309 | 0.0228 | 0.0157 | 0.0329 | 0.0171 | 0.0116 | 0.0318 | 0.0230 | 0.0166 |
| | | (S4) | | | (S5) | | | (S6) | | |
| | | $\pi_{\text{th}}=0$ | $\pi_{\text{th}}=0.1$ | $\pi_{\text{th}}=0.2$ | $\pi_{\text{th}}=0$ | $\pi_{\text{th}}=0.1$ | $\pi_{\text{th}}=0.2$ | $\pi_{\text{th}}=0$ | $\pi_{\text{th}}=0.1$ | $\pi_{\text{th}}=0.2$ |
| | Storey | 0.0380 | 0.0097 | 0.0013 | 0.0337 | 0.0249 | 0.0296 | 0.0254 | 0.0011 | 0.0005 |
| (C5) | Quantile | 0.0371 | 0.0288 | 0.0125 | 0.0435 | 0.0386 | 0.0378 | 0.0365 | 0.0089 | 0.0012 |
| | Fisher | 0.0159 | 0.0022 | 0.0002 | 0.0147 | 0.0046 | 0.0035 | 0.0089 | 0.0000 | 0.0000 |
| | Sum | 0.0272 | 0.0113 | 0.0025 | 0.0302 | 0.0173 | 0.0126 | 0.0210 | 0.0010 | 0.0000 |
| | Storey | 0.0416 | 0.0141 | 0.0050 | 0.0390 | 0.0287 | 0.0259 | 0.0418 | 0.0222 | 0.0145 |
| (C9) | Quantile | 0.0447 | 0.0309 | 0.0184 | 0.0454 | 0.0424 | 0.0369 | 0.0472 | 0.0422 | 0.0317 |
| | Fisher | 0.0182 | 0.0026 | 0.0008 | 0.0170 | 0.0064 | 0.0022 | 0.0179 | 0.0081 | 0.0022 |
| | Sum | 0.0332 | 0.0141 | 0.0055 | 0.0313 | 0.0206 | 0.0118 | 0.0344 | 0.0225 | 0.0105 |

Table S8: FDR computed from 2000 simulations at significance level $\alpha = 0.1$ using Storey's BH procedure with parameter $\gamma = 0.5$. For $K_0 = K/2$ of the data agents $\pi_k = \pi_{\text{th}}$ while for the remaining data agents $\pi_k = 1 - \pi_{\text{th}}$.

| | | (S1) | | | (S2) | | | (S3) | | |
| | | $\pi_{\text{th}}=0$ | $\pi_{\text{th}}=0.1$ | $\pi_{\text{th}}=0.2$ | $\pi_{\text{th}}=0$ | $\pi_{\text{th}}=0.1$ | $\pi_{\text{th}}=0.2$ | $\pi_{\text{th}}=0$ | $\pi_{\text{th}}=0.1$ | $\pi_{\text{th}}=0.2$ |
|---|---|---|---|---|---|---|---|---|---|---|
| | Storey | 0.1117 | 0.0881 | 0.0716 | 0.1171 | 0.0736 | 0.0538 | 0.1123 | 0.0860 | 0.0730 |
| (C5) | Quantile | 0.1197 | 0.1101 | 0.1086 | 0.1161 | 0.1154 | 0.1085 | 0.1080 | 0.1114 | 0.1028 |
| | Fisher | 0.0589 | 0.0479 | 0.0392 | 0.0624 | 0.0368 | 0.0201 | 0.0565 | 0.0464 | 0.0354 |
| | Sum | 0.0919 | 0.0815 | 0.0679 | 0.0952 | 0.0724 | 0.0526 | 0.0879 | 0.0805 | 0.0599 |
| | Storey | 0.1064 | 0.0708 | 0.0489 | 0.1105 | 0.0513 | 0.0230 | 0.1159 | 0.0639 | 0.0461 |
| (C9) | Quantile | 0.1120 | 0.1074 | 0.1050 | 0.1198 | 0.1051 | 0.0984 | 0.1128 | 0.1089 | 0.0975 |
| | Fisher | 0.0641 | 0.0281 | 0.0153 | 0.0643 | 0.0191 | 0.0074 | 0.0653 | 0.0293 | 0.0164 |
| | Sum | 0.0890 | 0.0658 | 0.0458 | 0.0965 | 0.0589 | 0.0347 | 0.0940 | 0.0632 | 0.0483 |
| | | (S4) | | | (S5) | | | (S6) | | |
| | | $\pi_{\text{th}}=0$ | $\pi_{\text{th}}=0.1$ | $\pi_{\text{th}}=0.2$ | $\pi_{\text{th}}=0$ | $\pi_{\text{th}}=0.1$ | $\pi_{\text{th}}=0.2$ | $\pi_{\text{th}}=0$ | $\pi_{\text{th}}=0.1$ | $\pi_{\text{th}}=0.2$ |
| | Storey | 0.1073 | 0.0256 | 0.0038 | 0.0799 | 0.0642 | 0.0599 | 0.0631 | 0.0041 | 0.0005 |
| (C5) | Quantile | 0.1112 | 0.0830 | 0.0519 | 0.1187 | 0.1036 | 0.1071 | 0.0861 | 0.0257 | 0.0054 |
| | Fisher | 0.0606 | 0.0105 | 0.0007 | 0.0621 | 0.0137 | 0.0086 | 0.0261 | 0.0000 | 0.0000 |
| | Sum | 0.0861 | 0.0371 | 0.0093 | 0.0898 | 0.0543 | 0.0359 | 0.0551 | 0.0049 | 0.0000 |
| | Storey | 0.1151 | 0.0341 | 0.0100 | 0.1143 | 0.0736 | 0.0575 | 0.1093 | 0.0625 | 0.0294 |
| (C9) | Quantile | 0.1169 | 0.0932 | 0.0641 | 0.1223 | 0.1094 | 0.1065 | 0.1207 | 0.1095 | 0.0969 |
| | Fisher | 0.0648 | 0.0089 | 0.0018 | 0.0634 | 0.0196 | 0.0059 | 0.0654 | 0.0283 | 0.0067 |
| | Sum | 0.0967 | 0.0424 | 0.0158 | 0.0941 | 0.0632 | 0.0392 | 0.0947 | 0.0694 | 0.0374 |

control with the BH procedure in Theorem 3, not with Storey's BH procedure. It will be an interesting line of further work to determine if any adaptive BH procedure can maintain FDR control, and also to find a non-trivial lower bound on the FDR when applying the BH procedure to the conformal data contamination p-values.

## S4.3  COLLABORATIVE DATA SHARING

In this section, we numerically evaluate the performance of the proposed data sharing procedure in terms of classification accuracy. This is done for all eight data contamination scenarios with (C9) the class-wise score based on PU classification using LR, and to solve the classification task we use the well-studied, simple, and robust method in SVC.

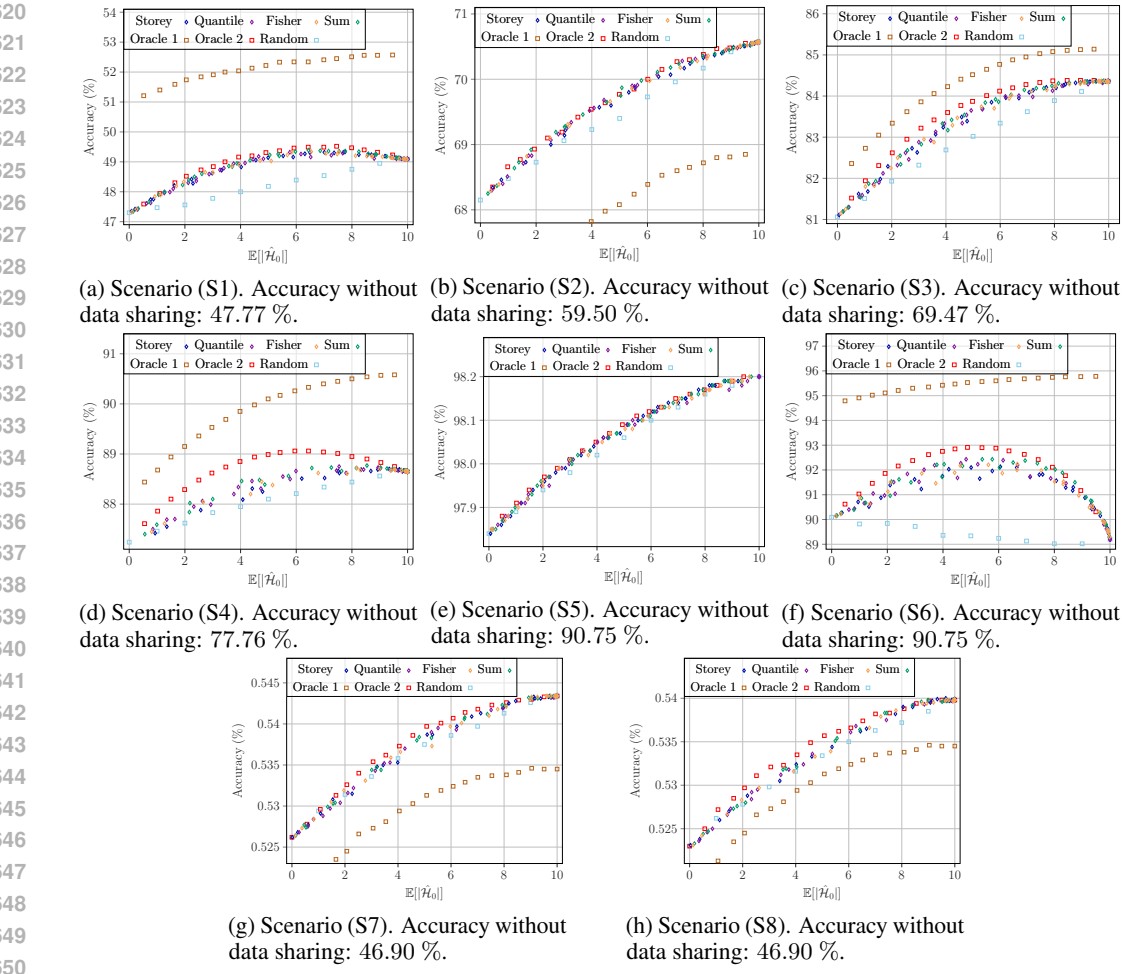

(a) Scenario (S1). Accuracy without data sharing: 47.77 %.

(b) Scenario (S2). Accuracy without data sharing: 59.50 %.

(c) Scenario (S3). Accuracy without data sharing: 69.47 %.

(d) Scenario (S4). Accuracy without data sharing: 77.76 %.

(e) Scenario (S5). Accuracy without data sharing: 90.75 %.

(f) Scenario (S6). Accuracy without data sharing: 90.75 %.

(g) Scenario (S7). Accuracy without data sharing: 46.90 %.

(h) Scenario (S8). Accuracy without data sharing: 46.90 %.

Figure S6: Plot of the model accuracy against the average number of collaborating data agents.

We show in Figure S6 the relation between the average number of collaborating data agents, $\mathbb{E}[|\hat{\mathcal{H}}_0|]$, and the average classification accuracy across 500 data simulations for each of the eight data contamination scenarios, with various hyperparameter settings $(\pi_{\mathrm{th}}, \alpha) \in \{0, 0.1, 0.2, 0.3, 0.4, 0.5, 0.6, 0.7\} \times \{0.05, 0.2, 0.5, 0.7\}$. No data subset selection is used meaning that all acquired datapoints are used for model fitting, since the focus of this work is on the proposed conformal data contamination tests. We observe that in scenarios (S1), (S3), (S4), and (S6) the oracles and the proposed methods outperform the *random* baseline meaning that in those scenarios a substantial boost in accuracy can be achieved by carefully selecting data agents for collaboration. Also, in these scenarios *Oracle 1* is clearly outperforming *Oracle 2* showing that there is further room for improvement by incorporating data subset selection. Meanwhile, in scenarios (S2), (S5), (S7), and (S8), i.e., the scenarios with optical coherence tomography images, feature noise on MNIST, and feature noise on CIFAR-10, it appears that the contaminated data is (nearly) as useful for subsequent classification model training as the inliers. In all the considered scenarios, the proposed data sharing procedures outperform the case of no data sharing, emphasizing the value of collaboration. Moreover, the proposed data sharing procedures reaches the performance of *oracle 2* in most cases, showing the efficiency of the proposed methodology. A notable exception is scenario (S4), i.e., the FEMNIST data contamination scenario, and we also observed in Table S3 that outlier detection was difficult in this scenario, hence, showing that in this case it would be beneficial to further improve the conformal score.

To give a comprehensive insight into the improvements in accuracy, we display in Figure S7 the empirical CDF of the improvement in accuracy compared to the *random* baseline when the average number of collaborating data agents is $\mathbb{E}[|\hat{\mathcal{H}}_0|] = 6$. The improvement in ac-

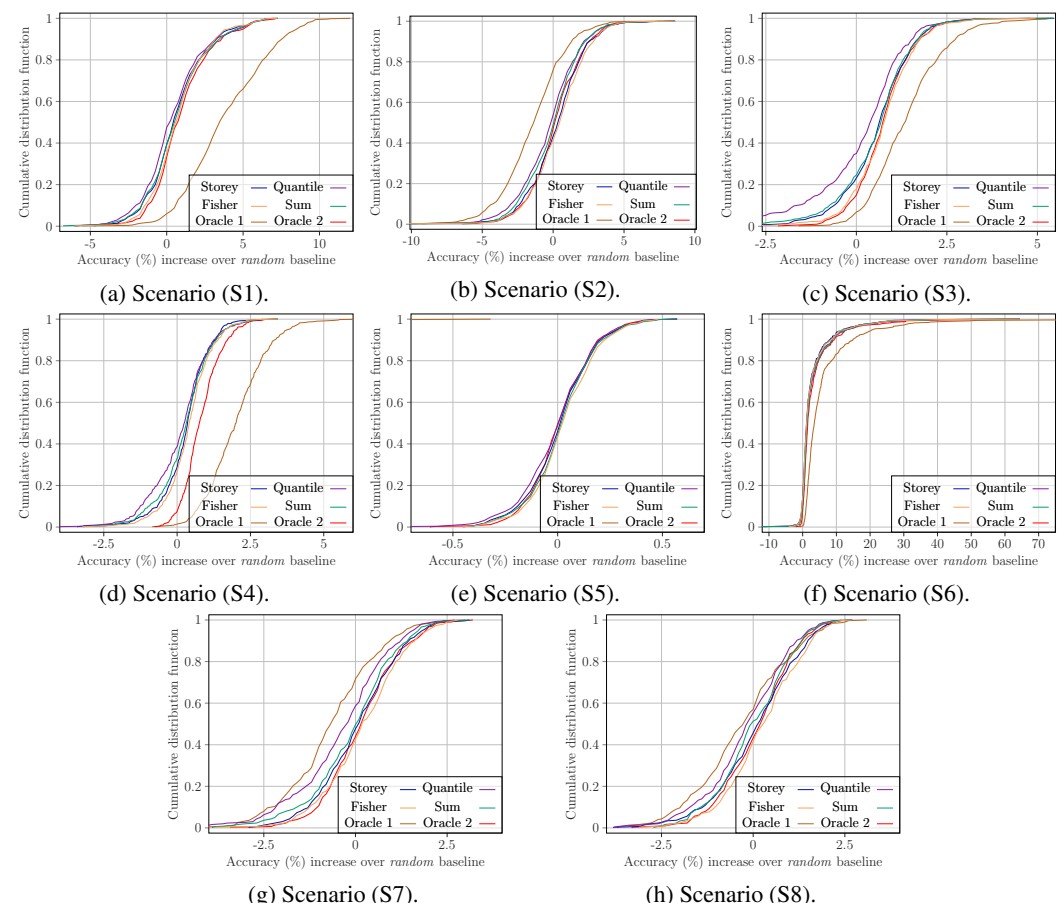

(a) Scenario (S1).  (b) Scenario (S2).  (c) Scenario (S3).

(d) Scenario (S4).  (e) Scenario (S5).  (f) Scenario (S6).

(g) Scenario (S7).  (h) Scenario (S8).

Figure S7: Plot of the empirical CDF of the difference in model accuracy compared to the *random* baseline when the average number of collaborating data agents is $\mathbb{E}[|\hat{\mathcal{H}}_0|] = 6$.

curacy compared to the *random* baseline is computed as $\text{Accuracy}_{\text{method}} - \text{Accuracy}_{\text{Random}}$ where $\text{Accuracy}_{\text{method}}$ is the accuracy in a given Monte Carlo simulation with $\text{method} \in \{\text{Random}, \text{Oracle1}, \text{Oracle2}, \text{Storey}, \text{Quantile}, \text{Fisher}, \text{Sum}\}$. Comparing the methods relative to the *random* baseline has the advantage of taking the randomness in the sampling of the data into account. If we compare the distribution of the accuracies directly a lot of the variation is explained from the variation in the information available from the sampled datasets.

Figure S7 illustrate, similarly to Figure S6, the gains in accuracy achieveable to carefully selecting data agents for collaboration in data contamination scenarios (S1), (S3), (S4), and (S6). For scenario (S1) we observe increases in accuracy of 5 percentage points in some simulations with the proposed methods, while for scenario (S6) we have increases in accuracy reaching 20 percentage points and even more in some simulations with the proposed methods. We are also interested in how often the proposed methods yield a higher accuracy than the *random* baseline which can be read from Figure S6, but we report it also in Table S9. Here we see that with the Fisher test we can improve in accuracy over the *random* baseline in 65 %, 83 %, 73 %, and 85 % of the simulations in data contamination scenarios (S1), (S3), (S4), and (S6), respectively.

### S4.4 SCALABILITY TO LARGE SCALE PROBLEMS

Here, we explore the performance of the data acquisition protocol for problems of larger scale. We consider three scenarios: full MNIST with label noise, FEMNIST with lower- and uppercase handwritten letters *a* through *j*, and full CIFAR-10 mixed with CIFAR-10C contaminated with brightness. All three scenarios have 10 classes, and we set $\ell = 300$, $n - \ell = 250$, and $m = 50$. Further, for MNIST and FEMNIST we set the number of data agents to be $K = 100$, while for

Table S9: Empirical probability to improve accuracy compared to the *random* baseline.

|      | Oracle 1 | Oracle 2 | Storey | Quantile | Fisher | Sum   |
|------|----------|----------|--------|----------|--------|-------|
| (S1) | 0.940    | 0.666    | 0.606  | 0.524    | 0.650  | 0.608 |
| (S2) | 0.258    | 0.514    | 0.506  | 0.412    | 0.536  | 0.480 |
| (S3) | 0.926    | 0.820    | 0.776  | 0.750    | 0.828  | 0.800 |
| (S4) | 0.986    | 0.924    | 0.712  | 0.614    | 0.734  | 0.670 |
| (S5) | 0.000    | 0.512    | 0.540  | 0.504    | 0.566  | 0.556 |
| (S6) | 0.996    | 0.924    | 0.858  | 0.876    | 0.854  | 0.892 |
| (S7) | 0.330    | 0.597    | 0.560  | 0.447    | 0.540  | 0.613 |
| (S8) | 0.447    | 0.603    | 0.577  | 0.447    | 0.510  | 0.627 |

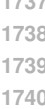

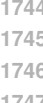
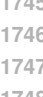
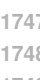
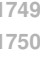

(a) FEMNIST. Accuracy without data sharing: 66.68 %.  (b) MNIST label noise. Accuracy without data sharing: 84.11 %.  (c) CIFAR-10C brightness. Accuracy without data sharing: 31.20 %.

Figure S8: Plot of the model accuracy against the average number of collaborating data agents.

the CIFAR-10 scenario $K = 50$ due to limited amount of total data. As previously, we use SVC as the classifier, $f^*$. For the MNIST and FEMNIST data we use the conformal score (C9) and for CIFAR-10 we use conformal score (C11). Results are only shown for the Storey and Fisher tests with hyperparameters $(\pi_{\text{th}}, \alpha) \in \{0, 0.1, 0.2, 0.3, 0.4, 0.5\} \times \{0.01, 0.1\}$ when averaged across 10 scenario simulations for MNIST and FEMNIST scenarios, and 50 for the CIFAR-10 scenario. Note that the variability in accuracy across data simulations is reduced in large scale problems.

The results are shows in Figure S8. In all three scenarios, substantial improvements in accuracy is achieved as compared to the *random* baseline, and in most cases the proposed methods are capable of (nearly) reaching the performance of *oracle 2*, as in the small scale problems previously considered. Generally, the performance gains of the proposed protocols, as compared to the *random* baseline, tends to be more pronounced here than in the small scale problems. This verifies that the proposed data acquisition protocol scales well to problems of larger scale which is an essential property for practical deployment.

### S4.5 DATA-DRIVEN HYPERPARAMETER SELECTION

For the proposed collaborative data sharing procedure there are a number of hyperparameters, see Section S2.1 for an overview. In this section, we will consider a data-driven approach to selecting key hyperparameters. We observed in Figure S6 for some of the data contamination scenarios a concave relation between the accuracy and $\mathbb{E}[|\hat{\mathcal{H}}_0|]$, and so in practice we may be interested in selecting the optimal hyperparameters in terms of accuracy.

The data-driven hyperparameter selection we propose here can be summarized as: (i) order the $K$ data agents in terms of contamination and determine $\hat{\mathcal{H}}_0$; (ii) fit the classification model on the data $\{Z_i\}_{i=1}^{\ell} \bigcup \{Z_i^k : i \in [m], k \in \hat{\mathcal{H}}_0\}$; (iii) evaluate the accuracy on the data $\{Z_i\}_{i=\ell+1}^{n}$ denoted $L_{\text{val}}(\pi_{\text{th}}, \alpha)$ noting that $\hat{\mathcal{H}}_0$ depends on the hyperparameters $\pi_{\text{th}}$ and $\alpha$; (iv) set the hyperparameter as $(\hat{\pi}_{\text{th}}, \hat{\alpha}) = \arg\max_{(\pi_{\text{th}}, \alpha) \in G} L_{\text{val}}(\pi_{\text{th}}, \alpha)$ where $G$ is a grid of hyperparameter values.

We compare with the *random* and *oracle 2* baselines: for the *random* baseline the hyperparameter is $K_{\text{budget}}$, and for baseline *oracle 2* the hyperparameter is $\pi_{\text{th}}$.

In Table S10, we show the mean and standard deviation of the accuracy (%) across 500 data simulations when using the data-driven hyperparameter selection. In all cases, we consider the

Table S10: Mean (standard deviation) of accuracy ($\%$) across $500$ data simulations with data-driven hyperparameter selection.

|      | Oracle 2      | No sharing    | Random        | Storey        | Quantile      | Fisher        | Sum           |
|------|---------------|---------------|---------------|---------------|---------------|---------------|---------------|
| (S1) | 48.64 (3.00)  | 47.77 (2.90)  | 47.96 (2.98)  | 48.55 (3.02)  | 48.77 (3.07)  | 48.56 (3.00)  | 48.25 (3.06)  |
| (S2) | 69.49 (3.22)  | 59.50 (5.42)  | 69.35 (3.27)  | 69.48 (3.34)  | 69.20 (3.33)  | 69.36 (3.27)  | 69.00 (3.34)  |
| (S3) | 83.96 (1.59)  | 69.47 (4.39)  | 83.75 (1.73)  | 83.03 (2.37)  | 83.08 (2.33)  | 83.09 (2.34)  | 82.42 (2.27)  |
| (S4) | 88.78 (1.47)  | 77.66 (3.11)  | 88.22 (1.53)  | 88.50 (1.56)  | 88.44 (1.62)  | 88.46 (1.62)  | 88.42 (1.65)  |
| (S5) | 98.02 (0.28)  | 90.75 (2.03)  | 97.97 (0.29)  | 97.99 (0.30)  | 98.01 (0.28)  | 98.01 (0.30)  | 97.91 (0.31)  |
| (S6) | 92.40 (4.01)  | 90.75 (2.03)  | 90.86 (5.09)  | 91.71 (4.58)  | 91.86 (4.67)  | 91.93 (4.36)  | 91.57 (4.87)  |

settings as described in Section S4 and use the conformal score (C9). First, we notice that for scenarios (S2)-(S5) all the methods are close in terms of accuracy, since in these scenarios the general rule is that more data is almost always better, as has also been noted above. On the other hand, for scenarios (S1) and (S6) notable gains in accuracy can be achieved with the proposed method. We further observe that we are not quite able to reach the peak accuracies observed in Figure 3 with the proposed methods, however, it is quite close. For this reason, we deem that the data-driven hyperparameter selection paves the way for a practical deployment of the procedure, and note that the optimization of other hyperparameters could also be done with the same approach. Notice also that for the *random* baseline, the mean accuracy surpasses the highest accuracy observed in Figure 3, due to the implicit dependency on the contamination factors in choosing $K_{\text{budget}}$.

This data-driven hyperparameter selection still lacks attention to the cost of collaboration. For instance, in scenarios (S2)-(S5), where more data is almost always better, the optimal choice is often to collaborate with all the data agents. In a specific application, the loss function could be defined to take the cost of collaboration into account.

## S5    USE OF LARGE LANGUAGE MODELS

Large language models have been used for the purpose of text editing.

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
