# OpenReview forum: "Conformal Data Contamination Tests for In-distribution Data Acquisition"
_ICLR.cc/2026/Conference — Submitted to ICLR 2026_

### Official Review · Reviewer_fRGL · 2025-10-31

**Soundness:** 3
**Presentation:** 2
**Contribution:** 2
**Rating:** 4
**Confidence:** 3

**Summary:**

The paper considers the problem that data buyers want to acquire similar personalised data and need quality guarantees prior to data acquisition. The paper proposes a distribution-free solution that selects data only from agents with less contaminated data (data from another distribution). The main contributions include new theoretical two-sample testing procedures, data sharing procedures and experiments on medical image datasets to validate the effectiveness and practicality.

**Strengths:**

1. The problem is generally well motivated in the introduction. It may be helpful to further explain why the data buyers can only purchase from few buyers and would not know the distribution or value of others’ data beforehand (e.g., by getting them to predict on a validation set).
2. The solution seems theoretically grounded.

**Weaknesses:**

1. The main paper or the appendix should provide more background for unfamiliar readers, e.g., on the BH procedure and the definition of PDRS.
2. Sec 3 and 4 should describe technical challenges involved and describe the significance/implications of the results.

**Questions:**

1. Can you provide some intuition on why the proposed testing procedures do not require distributional assumptions?
2. Is the method less efficient and effective on larger datasets, e.g., the full MNIST? Also, the experiments consider mislabeled data. Does it work when the data is correctly labeled but the class distribution differs?

---

> ### Author Response · Authors · 2025-11-19
>
> We thank the reviewer for the close reading and honest review of our paper. Below we give point-by-point replies.
>
> > It may be helpful to further explain why the data buyers can only purchase from few buyers and would not know the distribution or value of others’ data beforehand (e.g., by getting them to predict on a validation set).
>
> The setting of our paper is compatible both with few and many data agent scenarios. Scalability to many data agents is achieved through the use of the Benjamini-Hochberg procedure which is designed for large scale testing. However, in a practical scenario, communication constraints restricts a given data agent to interact with only a subset of other data agents per round, and this motivates the scenarios of the numerical experiments with just $K=10$ other data agents. Additionally, it is a strength of our proposal that information on the distribution of value of others' data is not required to be known beforehand, particularly one may encounter scenarios where other data agents falsify their data quality. Such issues are avoided in our setting by virtue of data sharing. Your proposal to predict on a validation set is valid and could perhaps be used to rank data agents in terms of distributional conformity, however, such a technique lacks the statistical guarantees of our proposal.
>
>
> > The main paper or the appendix should provide more background for unfamiliar readers, e.g., on the BH procedure and the definition of PDRS.
>
> The BH procedure is defined in Section 2.1 (lines 158-159) and details such as PRDS are relegated to the supplementary material, see Section S1.1 (lines 872-875). Further details regarding multiple testing theory is out of the scope of the paper and instead we refer the reviewer to Benjamini & Yekutieli (2001).
>
>
> > Sec 3 and 4 should describe technical challenges involved and describe the significance/implications of the results.
>
> A core motivator of the proposed procedure is the technical simplicity and ease of implementation, see lines 082-092. The implications of the results are described in the **"Main Contributions"** paragraph in Section 1 (lines 102-142). To summarize here, the significance/implications are both in terms of novel statistical tools that are general and so may have wide applications as well as its direct motivation through the collaborative learning scenario of the paper.
>
>
> > Can you provide some intuition on why the proposed testing procedures do not require distributional assumptions?
>
> The property of not requiring distributional assumptions is a strength and comes by virtue of design. The point of all conformal inference is to provide valid uncertainty quantification without making any distributional assumptions, other than assuming exchangeability of the data. With exchangeability, all conformal scores have the same chance of being the smallest (or i-th smallest). This allows to measure how likely it is that a given test score is too extreme to fulfil the null hypothesis of exchangeability. For further details on how conformal methodology can provide valid statistical inference without requiring distributional assumptions, we refer for example to Angelopoulos & Bates (2023).
>
>
> > Is the method less efficient and effective on larger datasets, e.g., the full MNIST?
>
> The method is not less efficient and effective on larger datasets; it depends on the specific data and the difficulty of the classification task solved by the model $f^*$. Generally speaking, the method actually tends to be more effective for larger datasets: if $n$ (and $\ell$) are larger, a better conformal score can be fitted; and when $m$ is larger, the data contamination tests increase in statistical power (see also lines 267-273 in Section 3 of the paper). However, note that if data agent 0 already has access to a very large dataset (when $n$ is very large), there is no need for collaboration.
>
>
> > Also, the experiments consider mislabeled data. Does it work when the data is correctly labeled but the class distribution differs?
>
> The method also works for class distribution differences. In Section 5, we consider two different data contamination scenarios, and we consider four additional data contamination scenarios in Section S4. The reviewer is correct that one of these data contamination scenarios is mislabeled data, however, the other five data contamination scenarios are examples of class distribution differences, and have correctly labeled data.
>
>
> ## References
>
> Benjamini & Yekutieli (2001), *The control of the false discovery rate in multiple testing under dependency*, The Annals of Statistics.
>
> Angelopoulos & Bates (2023), *Conformal Prediction: A Gentle Introduction*, Foundations and Trends in Machine Learning.
>
>
> ---
> We hope this reply answers your questions satisfactorily. We are available for further discussion.

---

> > ### Comment · Reviewer_fRGL · 2025-11-26
> >
> > Thank you for the reply! I find the additional experiments in Section S4 helpful. Additionally, I think it will benefit unfamiliar readers to reference the appendix in the main paper (e.g., S1.1 when PRDS is first introduced) and if the appendix contains necessary background knowledge from other papers.
> >
> > > The method is not less efficient and effective on larger datasets; it depends on the specific data and the difficulty of the classification task solved by the model $f^*$. Generally speaking, the method actually tends to be more effective for larger datasets: if $n$ (and $\ell$) are larger, a better conformal score can be fitted; and when $m$ is larger, the data contamination tests increase in statistical power (see also lines 267-273 in Section 3 of the paper).
> >
> > Can this be empirically shown? I observed that another reviewer has also asked about why the experiments were conducted with few agents and data points.

---

> > > ### Author Response · Authors · 2025-11-27
> > >
> > > We are grateful for your careful response, and state our reply below.
> > >
> > > > Additionally, I think it will benefit unfamiliar readers to reference the appendix in the main paper (e.g., S1.1 when PRDS is first introduced) and if the appendix contains necessary background knowledge from other papers.
> > >
> > > Per the reviewer's suggestion, we include in the main paper pointers to the background detailed in the supplementary material.
> > >
> > >
> > > > Can this be empirically shown? I observed that another reviewer has also asked about why the experiments were conducted with few agents and data points.
> > >
> > > We thank the reviewer for this question. We conducted an ablation study in Section S3.3 with a simple Gaussian data example, in which we tried varying the calibration data size ($n-\ell$) and the test data size ($m$). The results displayed in Figure S3 show our findings, herein, validating that the data contamination tests benefit from larger data sizes.
> > >
> > > In light of the reviewer's comment and that of Reviewer N3w4, we have numerically tested the proposed protocol in a scenario with many data points (approx. 10000) and many data agents (K=100). We will report the findings in the supplementary of the revised manuscript: in summary, as we hypothesized, the results are comparable to that of the previous analysis.

---

### Official Review · Reviewer_iwQx · 2025-10-31

**Soundness:** 3
**Presentation:** 3
**Contribution:** 3
**Rating:** 6
**Confidence:** 3

**Summary:**

The paper introduces conformal data contamination tests. These tests are distribution-free, allowing a data buyer to check multiple outside data sources and retain only those whose data are not "too contaminated" relative to the buyer's own distribution.
For each candidate source, it builds conformal p‑values from a small preview batch, combines them into a single p‑value, and then applies Benjamini-Hochberg across sources to control FDR.
Authors demonstrate that in medical dataset/MNIST image classification experiments, the procedure can select better collaborators and improve downstream accuracy.

**Strengths:**

1. The paper is well written and easy to follow.

2. The idea behind the paper is simple yet effective, and I think it addresses an important problem.

3. The paper provides both rigorous theoretical results and experimental evaluation (although see weaknesses).

**Weaknesses:**

Overall, I like this paper. I only have a concern about the choice of datasets for experiments and the contamination procedure. The authors considered hand-crafted noise (e.g., label noise), but not a real-world type of noise. One way to address this could be by considering additional datasets that are designed for it. It could be CIFAR-10C or CIFAR-100C, but also, for example, ImageNet (clear) vs. ImageNet-R.
In my opinion, this would be a more realistic approach.

Additionally, I have a conceptual question about the whole approach (this is not necessarily a weakness). The approach assumes that each external agent can reveal $m$ samples for demonstration in every round, which may be problematic in privacy-sensitive settings. Therefore, each of the agents may utilize watermarks / other curruptions to preserve privacy. And these corruptions may differ from one agent to another. What do you think could be done in this case?

**Questions:**

See Weaknesses.

---

> ### Author Response · Authors · 2025-11-19
>
> Thank you for the expert review and for acknowledging the merits of our work. We hope that you find our reply to your specific questions, as presented below, satisfactory.
>
> > The authors considered hand-crafted noise (e.g., label noise), but not a real-world type of noise. One way to address this could be by considering additional datasets that are designed for it. It could be CIFAR-10C or CIFAR-100C, but also, for example, ImageNet (clear) vs. ImageNet-R. In my opinion, this would be a more realistic approach.
>
> In the main paper, we consider in Section 5 a real-world type of contamination in the case of retinaMNIST compared to EyePACS. We further consider two other types of real-world data contamination with medical datasets in Section S4. These data examples are, in our opinion, closer to a real-world type of noise, and more relevant to the motivating example of hospitals given in the introduction, than the artificial contamination in, for instance, CIFAR-10C. Still, we appreciate the suggestion of the reviewer to explore the potential of our methodology on additional datasets, for instance CIFAR-10C, CIFAR-100C, and ImageNet-R. We are currently conducting numerical experiments with the CIFAR-10 and CIFAR-10C datasets.
>
>
> > The approach assumes that each external agent can reveal samples for demonstration in every round, which may be problematic in privacy-sensitive settings. Therefore, each of the agents may utilize watermarks / other curruptions to preserve privacy. And these corruptions may differ from one agent to another. What do you think could be done in this case?
>
> We agree that our setting is not privacy-aware; the focus of this work is to enable quality data acquisition, similar to Blum et al. (2017), Huang et al. (2023), Lu et al. (2024), and Ananthakrishnan et al. (2024), however, under data contamination scenarios. Had data privacy been required, the practical solution would have been related to federated learning. There are some works on personalized federated learning based on forming coalitions and training a personalized federated model for each participating coalition, however, there are plenty of downsides of federated learning, for instance sensitivity to model poisoning attacks. Such issues are avoided in a data sharing paradigm, as each individual data agent has full control over the data used for model training. To preserve some degree of privacy in a data sharing framework, one approach is to transform each datum by a non-invertible mapping to a latent space, and perform all the machine learning pipeline in this latent space. An example of such a transformation could be based on principal component analysis. This is an interesting line of research and will be the topic of future works.
>
> Along this discussion, we also find it relevant to mention a line of work dealing with federated conformal prediction. This work differs from ours fundamentally as it deals with uncertainty quantification in the downstream classification task in the federated setting. Connected to the related works discussion mentioned in the reply to reviewer EQ8J, we add the following:
>
> *If data agents have strict privacy constraints, federated conformal methodologies can be used for distributed uncertainty quantification. Conformal inference uses the empirical quantile of the calibration scores, and so Humbert et al. (2023) proposed to use the quantile-of-quantiles among data agents, while Lu et al. (2023) proposed a federated quantile estimator. Concurrently, Plassier et al. (2023) developed a technique based on weighted conformal prediction and federated quantile estimation, which they further improved in Plassier et al. (2024). Personalization was considered in Min et al. (2025) who proposed to use the data of external agents to learn the conformal score using the idea of localized conformal prediction. Contrary to this paper, these aforementioned works deal with uncertainty quantification in the downstream classification task.*
>
>
> ## References
>
>
> Blum et al. (2017), *Collaborative PAC Learning*, NIPS.
>
> Huang et al. (2023), *Evaluating and Incentivizing Diverse Data Contributions in Collaborative Learning*, FL-ICML.
>
> Lu et al. (2024), *Data Acquisition via Experimental Design for Data Markets*, NIPS.
>
> Ananthakrishnan et al. (2024), *Delegating Data Collection in Decentralized Machine Learning*, AISTATS.
>
> Humbert et al. (2023), *One-Shot Federated Conformal Prediction*, ICML.
>
> Lu et al. (2023), *Federated Conformal Predictors for Distributed Uncertainty Quantification*, ICML.
>
> Plassier et al. (2023), *Conformal Prediction for Federated Uncertainty Quantification Under Label Shift*, ICML.
>
> Plassier et al. (2024), *Efficient Conformal Prediction under Data Heterogeneity*, AISTATS.
>
> Min et al. (2025), *Personalized Federated Conformal Prediction with Localization*, NIPS.
>
>
> ---
> We are available for discussion of any further questions or doubts you may have regarding our work.

---

> > ### Comment · Reviewer_iwQx · 2025-11-20
> >
> > Dear authors,
> >
> > Thank you for your detailed reply.
> >
> > > We are currently conducting numerical experiments with the CIFAR-10 and CIFAR-10C datasets.
> >
> > Thank you! I think this will strengthen the paper and make experimental evidence more convincing.
> >
> >
> > > our setting is not privacy-aware; we also find it relevant to mention a line of work dealing with federated conformal prediction;
> >
> > It is also important to emphasize this point in the paper to better frame the contribution, as well as add the discussion on the federated conformal prediction paper in the "Related Work" section.
> >
> > ----
> >
> > Overall, I remain positive about the paper, and I am willing to increase my evaluation score to Accept once the changes are incorporated.

---

### Official Review · Reviewer_N3w4 · 2025-10-31

**Soundness:** 3
**Presentation:** 2
**Contribution:** 2
**Rating:** 4
**Confidence:** 3

**Summary:**

This paper studies data sharing in the context of collaborative learning, where data owners have samples drawn from distributions $P_k$, which are Huber-contaminated versions of a common $P_0$. The authors design a conformal multiple testing procedure for agents to select a subset of collaborators whose contamination coefficient $\pi_k$ stands below a threshold. From a theoretical point of view, the paper proposes four valid p values, and show that the Storey p values are PRDS, hence making them compatible with a BH procedure. From a pratical point of view, the proposed procedure is implemented on MNIST and RetinaMNIST/EyesPACS with 10 agents and 100 data points. The CDF of the different p values are displayed, as well as the accuracy as a funciton of the expected number of collaborators.

**Strengths:**

1) The idea of using hypothesis testing in collaborative learning to limit the harmful effects of data heterogeneity is interesting.
2) The statistical analysis is well conducted, with four new p values specifically tailor to test whether a distribution is contamined beyond a given threshold.
3) The experiments, even though the number of agents and data points are low, are fairly convincing.

**Weaknesses:**

1) Important related works have been neglicted. The idea of performing a statistical inference or test prior to collaboration in collaborative learning with agents having data of varying quality is not new. In particular, [1] already studies this problem and propose a similar solution to the one presented in this paper (estimating the discrepancy from $P_0$ and conditioning the collaboration on it). Likewise, [2] studies the selection of clients prior to collaborating as a bilevel problem. A discussion of these papers (among others) is missing.
2) A discussion about the four p-values introduced by the paper is missing. When is it better to use one rather than the others? Is there one of them that is easier to compute than others? It seems that the statistical analysis is a bit ad-hoc, and does connect well to the rest of the paper (about data sharing).
3) The other never discuss the complexity of their method, so it is not clear whether it is actually implementable or if its cost is probihitive in real-world setting (high dimension, a lot of data points and agents...)

[1] Capitaine, A., Boursier, E., Scheid, A., Moulines, E., Jordan, M., El-Mhamdi, E. M., & Durmus, A. (2024). Unravelling in collaborative learning. Advances in Neural Information Processing Systems, 37, 97231-97260.

[2] Hashemi, D., He, L., & Jaggi, M. (2024). Cobo: Collaborative learning via bilevel optimization. Advances in Neural Information Processing Systems, 37, 15550-15574.

**Questions:**

1) Can you compare your problem and method to references [1] and [2]?
2) Can you further discuss the four p values introduced in theorem 1 and 2? In particular, is there one that should be favored from a practical point of view (I suspect "Storey" given theorem 3). In this case, what is the interest of the three others?
3) Can you discuss the complexity of your method? Is the computational / time cost of implementation the main reason why your experiments were conducted with few agents and data points?

---

> ### Author Response · Authors · 2025-11-19
>
> Thank you for your thoughtful review and constructive feedback. Below we answer your questions point-by-point.
>
> > Can you compare your problem and method to references [1] and [2]?
>
> The work [1] mentioned by the referee is a very interesting theoretical contribution that deals with a problem that is different from our problem setting. Perhaps the most important difference is that they deal with clients falsifying their data quality (assuming data quality is private), which is what causes the *"unravelling"* of the collaborative learning. Such an issue is not relevant within the data sharing framework of the current paper. Further, their statistical framework differs from ours in multiple ways: (i) they consider empirical risk minimization, and (ii) their assumption **H2** is incompatible with our setting. Precisely, they assume that each agent has access from data with a distribution different from $P_0$ while we assume that our agents access to data with distribution $P_0$, eventually up to some contamination. Both settings have their merits/applications and a full discussion of the connection goes way beyond our article or [1].
>
> The reference [2] proposes a federated learning technique, fitting personalized models for each contributing data agent, hence, their work is not about data acquisition which is the topic of the current paper. In a scenario where privacy conserns are central, techniques such as the proposal in [2] are relevant, however, if the data agents do not have strict privacy constraints, data sharing procedures are more effective. Among the downsides of federated learning are the costs in terms of communication and computation, and the sensitive to model/data poisoning. See also our motivation of the data sharing framework in lines 059-066 in the Section 1.
>
>
> > Can you further discuss the four p-values introduced in Theorems 1 and 2? In particular, is there one that should be favored from a practical point of view (I suspect "Storey" given Theorem 3). In this case, what is the interest of the three others?
>
> Based on the extensive numerical experiments presented in Section 5, Section S3, and Section S4, we describe at the end of Section 5.2 (lines 451-457) our recommendations regarding which of the four p-values to use.
>
>
> > Can you discuss the complexity of your method? Is the computational/time cost of implementation the main reason why your experiments were conducted with few agents and data points?
>
> One of the strengths of our proposed method is its low complexity. Note that computing a conformal data contamination test p-value as defined in Eqs. (3)-(6) requires at most $m$ evaluations of well-known CDFs (for Quantile and Storey the CDF of a negative hypergeometric distribution, for Fisher the CDF of a chi-squared distribution, and for Sum the CDF of an Erlang distribution). The more computationally complex part of the proposed method is the fitting of the conformal score function $\hat{s}$, which notably depends on the choice of the practitioner: the methodology complies with a wide range of conformal scores, see Section S4 (lines 1256-1267) for a comprehensive study with various conformal score functions. In the numerical experiments of Section 5, the conformal score function is defined through a binary classifier, specifically penalized logistic regression. Clearly, fitting a penalized logistic regression model does not present a substantial computational constraint. The most computationally complex part of the framework is fitting the classifier $f^*$, which is unrelated to the proposed methodology, and by virtue of our method is only required to be done once. Since the computational complexity of the proposed methodology is negligble we have not emphasized this aspect in the paper. We have not conducted experiments with a large number of data agents and huge datasets since this would not provide any new insights into the proposed methodology, and so would only serve to slow down the scientific process.
>
> For the setting of Section 5 on the EyePACS data with a laptop equipped with 11th Gen Intel(R) Core(TM) i7-1165G7 @ 2.80GHz, the method takes on average approx. 0.08 seconds (approx. 0.06 seconds for conformal score fitting and  approx. 0.02 seconds for conformal data contamination test p-value computation).
>
> ---
> We expect this reply clears up your doubts regarding our paper. In any case, we are available for further discussions.

---

> > ### Comment · Reviewer_N3w4 · 2025-11-25
> >
> > I thank the authors for their thorough discussion.
> >
> > - **Regarding related literature**: I thank the author for the detail comparison with [1] (although I am not entirely convinced by the difference they highlight regarding $P_0$) and [2]. My overall feeling is that the literature about statistical setting has been carefully covered, but the literature about data sharing with heterogeneous agents has been somehow neglected. In my opinion, the introduction in the revised version should include a "related work" section discussing [1] (maybe [2]) along with other relevant papers about data acquisition with strategic/heterogeneous agents (there are many!).
> >
> > - **Regarding the discussion about the p-values introduced in the paper**: While the discussion following from the experiments in lines 451-457 is interesting, it cannot replace a proper theoretical discussion of the p-values after the statements of theorem 1, 2 and 3. In my opinion, these p-value should be carefully justified and motivated (all the more as thre are four of them!): what are the theoretical pros and cons of each of them? How do they relate? What is the practical relevance of each p-value for data acquisition? I feel such a discussion would significantly improve the flow of the paper.
> >
> > - **Regarding complexity and experiments**: I thank the authors for their convincing discussion about complexity, which should be included in the revised version in my opinion.
> > However, I cannot agree with the following statement: ``We have not conducted experiments with a large number of data agents and huge datasets since this would not provide any new insights into the proposed methodology, and so would only serve to slow down the scientific process."" Studying how methods scale with the number of data points (both from a convergence and computational point of view) is of prime importance nowadays. Conducting an experiment with 100 data points is very far from the current scientific standards, particularly in a conference like ICLR! I think that the experiments should be conducted with more data points to be truly convincing.

---

> > > ### Author Response · Authors · 2025-11-26
> > >
> > > We thank the reviewer for the detailed and timely engagement in discussions.
> > >
> > > - **Regarding related literature**: In the revised version, we will include in the Introduction a paragraph of related works to cover literature on data acquisition with heterogeneous data agents, including reference [1] as well as those mentioned in our rebuttal to reviewer iwQx [Huang et al. (2023); Lu et al. (2024); Ananthakrishnan et al. (2024)], amongst others.
> > >
> > > - **Regarding discussion about the p-values**: In the revised version, we will include in Section 3 a paragraph discussing the theoretical and practical relevance of the p-values, as you suggest.
> > >
> > > - **Regarding complexity and experiments**: As suggested, we will add the discussion on the complexity in the revised version. Also, we naturally agree with the reviewer that aspects of scalability are highly relevant for training, however, we emphasize that the proposed data acquisition protocol precedes training, so our hypothesis was that the "trend" will be the same. Nevertheless, two reviewers have raised valid arguments about verifying it, and we can now see that there is a need to verify it quantitatively. Hence, we have conducted numerical experiments with many data points (approx. 10000) and data agents (K=100). We observe a similar trend, leading to similar analysis as we made earlier, and we will include this as additional numerical results in the supplementary material.

---

### Official Review · Reviewer_EQ8J · 2025-11-01

**Soundness:** 3
**Presentation:** 2
**Contribution:** 2
**Rating:** 4
**Confidence:** 3

**Summary:**

In the setting where multiple data owners each have their own distinct dataset and where there is risk of data contamination, this paper proposes an approach toward providing some quality assurance on data sharing which builds on prior methods for conformal outlier detection. The proposed “conformal data contamination tests” improve on prior data data contamination tests by providing distribution-free validity (under some standard IID/exchangeability assumptions) rather than requiring parametric assumptions. To control issues with multiple-testing for the multiple data-sharing agents, the authors use the Benjamini-Hochberg procedure. They provide empirical evaluations across different collaborative learning settings to demonstrate robustness and effectiveness.

**Strengths:**

Overall the paper seems sound and well-motivated for the stated goals. That is, the proposed methods could have practical use in the setting of quality-assurance/contamination testing in collaborative data sharing. Relative to the prior work on conformal outlier detection by Bates et al. (2023)--where that work can be viewed as covering the special case where one wishes to test the contamination level of 0%, among other contributions--the current work provides seemingly valid hypothesis tests for any other contamination level. It is good that the authors address the multiplicity issues inherent to testing over multiple data-sharing agents, and the Benjamini-Hochberg procedure is reasonable for doing so.

**Weaknesses:**

**Novelty:** Although the proposed methods are well-motivated and could be useful for the setting studied, it currently does not seem to me that there is significant enough technical innovation in this paper to be of particular interest to the ICLR audience. While I appreciate that the prior conformal outlier detection methods in Bates et al (2023) do not cover the case of data contamination (to my understanding), conformal outlier detection under data contamination is studied by the ICML 2025 paper Bashari et al. (2025), “Robust conformal outlier detection under contaminated reference data,” which is not referenced in the current paper. The authors should add some discussion about how their work relates to that of Bashari et al. (2025). Beyond this reference, I’m wondering if this paper would be a better fit for a somewhat more specialized conference than ICLR, such as *Conformal and probabilistic prediction with applications* (COPA), or a journal focused on soundness, such as *TMLR*.

**Questions:**

Can the authors please clarify how their proposed work relates to that of Bashari et al. (2025)?

Bashari, M., Sesia, M., & Romano, Y. (2025). Robust conformal outlier detection under contaminated reference data. ICML.

---

> ### Author Response · Authors · 2025-11-19
>
> Thank you for your detailed review. We appreciate the effort and constructive feedback, and hope that the explanations below answer your questions.
>
> ## Novelty
> The reviewer is right that Bates et al. (2023) does not consider the case of data contamination. Although Bashari et al. (2025) also deal with conformal outlier detection, our work differs from it in two key aspects.
>
> The first is the purpose of the study. Contrary to our work, the purpose of Bashari et al. (2025) is to handle a scenario where the calibration data is *"contaminated"* by outliers, see for example their equation (3), and so they propose a technique to yield *robust* conformal prediction even with the presence of such *"contamination"*. Note that in our paper, the calibration data is not contaminated, rather we refer to *"contamination"* in the data from other data agents, which in the context of the conformal methodology is the test data. Further, the *"budget"* they mention is the labeling budget of the calibration data, and is not related to the *"budget"* mentioned in our article which deals with the possibility of acquiring new datasets in a data market scenario (Fernandez, 2020).
>
> The second is the analytical framework. Bashari et al. (2025) focus on controlling the type I error of a null hypothesis in a single testing scenario, see for example their equation (1). On the contrary, we consider a multiple testing scenario, where we propose novel tests, and we need to consider the FDR. Controlling the FDR in this setting requires the original results that are presented in our Section 3 with further details in Section S1.
>
> There is a line of work related to Bashari et al. (2025) dealing with *"corrupted"* (or *"contaminated*") calibration data for which we add the following discussion in the revised manuscript:
>
> *A related but different line of work in conformal inference deals with *corruption* in the calibration data, which in the general sense means that the calibration data is not exactly distributed according to the null distribution $P_0$. Such *corruption* violates the fundamental exchangeability assumption that conformal inference relies on, however, approximately valid inference is still possible using the idea of weighted conformal prediction from Tibshirani et al. (2019). The methodologies in this line of work is relevant for downstream uncertainty quantification following (imperfect) in-distribution data acquisition.*
>
> *The original work of Tibshirani et al. (2019) dealt with covariate shift. Subsequently, weighted conformal prediction was considered for label shift in Podkopaev & Ramdas (2021) and for generalized covariate shift and posterior drift in Wang & Qiao (2025). A detailed theoretical analysis of conformal prediction beyond exchangeability was conducted in Barber et al. (2023). Recently, *robust* conformal prediction has been considered: Bashari et al. (2025) proposed a technique to avoid *corruption* by detecting outliers in the calibration data, and Feldman & Romano (2024) considered scenarios with access to privileged information (not available at test time) that explains the distribution shift.*
>
>
> ## Choice of venue
> The ICLR conference clearly mention *"uncertainty quantification"* as a topic, which we cover, and in the past conformal methodology has been appreciated by the ICLR community, see for instance Wu et al. (2025), Plassier et al. (2025), amongst others. The COPA venue would also be suitable, but we do not see why it should be more suitable than ICLR. We do not believe that the theoretical guarantees we are providing means that we should focus on a journal in terms of soundness, both because ICLR as a venue also promotes soundness and we observe significant performance improvements for specific learning tasks.
>
>
> ## References
>
> Bashari et al. (2025), *Robust Conformal Outlier Detection Under Contaminated Reference Data*, ICML.
>
> Fernandez (2023), *Data-Sharing Markets: Model, Protocol, and Algorithms to Incentivize the Formation of Data-Sharing Consortia*, Proc. ACM Manag. Data.
>
> Tibshirani et al. (2019), *Conformal Prediction Under Covariate Shift*, NeurIPS.
>
> Podkopaev & Ramdas (2021), *Distribution-Free Uncertainty Quantification for Classification Under Label Shift*, UAI.
>
> Barber et al. (2023), *Conformal Prediction Beyond Exchangeability*, The Annals of Statistics.
>
> Sesia et al. (2024), *Adaptive Conformal Classification With Noisy Labels*, JRSSB.
>
> Feldman & Romano (2024), *Robust Conformal Prediction Using Privileged Information*, NeurIPS.
>
> Wang & Qiao (2025), *Conformal Prediction Under Generalized Covariate Shift with Posterior Drift*, AISTATS.
>
> Wu et al. (2025), *Conditional Testing based on Localized Conformal p-values*, ICLR.
>
> Plassier et al. (2025), *Probabilistic Conformal Prediction with Approximate Conditional Validity*, ICLR.
>
> ---
> We hope the details and clarifications given in this rebuttal aids in communicating the significance and originality of the paper.

---

### Author Response · Authors · 2025-12-02
**Author final remarks**

We thank all reviewers for their thoughtful comments. In the discussion with the reviewers, the feedback was centered on three main topics. Below, we summarize the discussion and outline the revisions made to the manuscript.


**Related works:**
Reviewers N3w4 and EQ8J asked us to compare our work to [1], [2], and Bashari et al. (2025). We have clarified that our work is clearly distinct from, but related to, these previous works. In light of these comments, we have improved the positioning of our paper within the literature by including a paragraph **Other related works** in the **Introduction** of the revised manuscript, which includes the references [1] and Bashari et al. (2025), amongst others.


**Real-world data contamination:**
Reviewer iwQx suggested to conduct numerical experiments with, for instance, CIFAR-10C to test our method on a *"real-world type of noise"*. While we do not agree that CIFAR-10C represents a more realistic type of noise than the medical datasets included in the submitted manuscript, we have nevertheless performed numerical experiments with CIFAR-10C. These results, included in the supplementary material of the revised manuscript, are consistent with our previous conclusions.


**Complexity and scalability:**
Reviewers N3w4 and fRGL expressed concerns regarding the complexity and the scalability to larger datasets of our proposed procedure, since the numerical experiments in the submitted manuscript were restricted to relatively small datasets. This impression was due to a misunderstanding, as our procedure precedes the actual training and is therefore computationally light, as well as scalable to a large number of data agents. Furthermore, we have conducted additional numerical experiments with larger datasets (and more data agents), and we observe results that are as convincing as, or even more convincing than, previous. These results are found in the supplementary material of the revised manuscript, and numerically validate the scalability of our proposed procedure.


**Other changes:**
- Reviewer iwQx mentioned the aspect of data privacy, something which our method is not designed to provide. To clarify this, we have included a **Limitations** paragraph in the main paper, noting therein that our method is not privacy-aware.
- Reviewer N3w4 had questions regarding which of the four introduced conformal data contamination p-values to use in a given setting. In the submitted manuscript, this was discussed in relation to the numerical experiments; however, per the reviewers' suggestion, we have expanded and moved the discussion to a paragraph **Selecting the test** in the **Main Results** section of the revised manuscript.

---
Following our replies, the reviewers have acknowledged the convincing answers given to all the posted questions and accepted the merits and contributions of this work. Reviewer iwQx have even explicitly stated their willingness to increase scores once the aforementioned changes were implemented in the revised manuscript.

---

### Meta-Review · Area_Chair_vYTG · 2025-12-25

**Summary:**

This paper proposes multiple hypothesis testing for the scenario of data sharing across agents where agents' data may be contaminated (OOD) with varying levels (mixture components). The authors propose a distribution free conformal testing procedure, and account for the challenge of multiple testing and multiple agents using the Benjamini-Hochberg procedure. Empirical evaluations suggest good data-acquisition procedure.

Reviewers comments/concerns:
1. Reviewers mentioned the need for better contextualization with prior work, especially recent works such as Bashari et al. (2025), Ananthakrishnan et al (2024), Capitaine, A., et al. (2024), Hashemi et al (2024).
2. A primary concern was the scale of empirical testing, number of agents and number of data points was considered too low
3. Request to add a more thorough discussion on computational complexity
4. A more detailed discussion on a more prescriptive and grounded discussion on choosing one of the fourconformal data contamination p-values for a given setting
5. Some reviewers asked for adding background information for readers unfamiliar with the field
6. Minor question related to setup, such as data-sharing under privacy constraints
7. Suitability to ICLR

**Reviewer Concerns:**

1. The authors did a commendable job earnestly answering reviewer responses.
2. Authors also are adding more experimentation (for 10000 data points and 100 agents) to confirm their method scales.
3. Authors have expanded the discussion of selecting the test
4. Added discussion on complexity in the response

**Reviewer Scores:**

1. I hypothesize the N3w4 might have increased scores from 4 to 6.
2. EQ8J took issue with suitabilty of the work to ICLR based on the scale of the experimentation, which the authors have attempted to address sincerely. I don't particularly agree with the idea that the paper is better suited for TMLR and/or a more focused venue. However, it is hard to contemplate whether EQ8J would have increased their score due to lack of engagement.
3. Authors have also addressed fRGL's questions well but the reviewer acknowledged they may be less familiar with the content, even so challenging to assess whether they would have increased their scores.

Overall this paper is currently very borderline according to these, not due to shortcomings in scientific contribution, but due to genuine uncertainty about reviewer discussion and potential scores. As such going by this uncertainty and leaning pessimistic, I recommend a reject, noting that this paper was one of the hardest to decide on in my pool of papers.

I urge the authors to restructure the organization of the draft a bit to incorporate more background, which the updated manuscript reflects to some extent.

---

### Decision · Program_Chairs · 2026-01-26

Reject